

# Enhanced isotopic approach combined with microbiological analyses for more precise distinction of various N-transformation processes in contaminated aquifer – a groundwater incubation study

**Sushmita Deb[1], Mikk Espenberg[2], Reinhard Well[3], , Michał Bucha[1], Marta Jakubiak[1], Ülo Mander[2], Dominika Lewicka-Szczebak[1]**

[1] Institute of Geological Sciences, University of Wrocław, Poland

[2] Institute of Ecology and Earth Sciences, University of Tartu, Estonia

[3] Thünen Institute of Climate-Smart Agriculture, Braunschweig, Germany

Correspondence to: Sushmita Deb (sushmita.deb@uwr.edu.pl)

**Abstract**

This study explores nitrogen transformations in groundwater from an agricultural area utilizing organic fertilizer (wastewaters from yeast production) integrating isotopic analysis, microbial gene abundance, and the FRAME (isotope FRactionation And Mixing Evaluation) model to trace and quantify nitrogen cycling pathways. Groundwater samples with elevated nitrate concentrations were subjected to controlled laboratory incubations with application of a novel low-level $^{15}N$ tracing strategy, to investigate microbial processes. Isotope analyses of nitrate, nitrite and nitrous oxide ($N_2O$), coupled with microbial gene quantification via quantitative PCR (qPCR), revealed a shift from archaeal-driven nitrification to bacterial denitrification in post-incubation suboxic conditions, stimulated by glucose addition. FRAME modeling further identified bacterial denitrification (bD) as the dominant pathway of $N_2O$ production, which was supported by increased *nosZI, nirK and nirS* gene abundance and observed isotope effects. Simultaneously to the intensive nitrate reduction, it was observed that the majority of nitrite is likely produced through nitrification processes linked to dissolved organic nitrogen (DON) oxidation. Nitrate reduction had minor contribution in the total nitrite pool. The results demonstrate the efficacy of integrating multi-compound isotope studies and microbial analyses to unravel nitrogen cycling mechanisms. This approach provides a robust framework for addressing nitrogen pollution in groundwater systems and improving water quality management strategies.

**Keywords**

Denitrification pathways, $\delta^{15}N$ and $\delta^{18}O$ isotopes, nitrate, nitrite, $N_2O$, microbial gene abundance, FRAME modeling, agricultural pollution,

**Introduction**

Nitrogen (N) is an essential nutrient for plant growth and global food production, forming a key component of nucleic acids and proteins. Although synthetic N fertilizers containing nitrate ($NO_3^-$) and/or ammonium ($NH_4^+$), have greatly influenced agricultural yields, their excessive use has significantly disrupted the N cycle, leading to $NO_3^-$ leaching in groundwater, emission of ammonia ($NH_3$), and gaseous forms of nitrogen oxides (nitric oxide (NO), nitrous oxide ($N_2O$), nitrogen dioxide ($NO_2$) which are of environmental concern (Sainju et al., 2020). These issues contribute to eutrophication of lakes, groundwater quality degradation, and greenhouse gas emissions, with $N_2O$ intensifying global warming and ozone depletion (Butterbach-Bahl et al., 2013). Controlling





NO$_3^-$ levels in aquatic systems presents substantial environmental challenges, particularly in
groundwater, due to the complexity of differentiating between its anthropogenic sources and
natural processes. Hence, the better understanding of N cycling is crucial to develop effective
solutions of environmental problems (Rütting et al., 2018).
Diverse microbial communities, including nitrogen-fixing bacteria, archaea, anammox bacteria,
nitrifiers, and denitrifiers, drive key N transformations, regulating its availability and mobility in
ecosystems. N undergoes complex transformation processes like nitrification, denitrification,
anammox, mineralization and immobilization (Deb et al., 2024) which regulate N availability to
plants and influence its movement in agricultural and natural systems. Biological fixation converts
atmospheric nitrogen (N$_2$) into bioavailable forms, while nitrification involves the microbial
oxidation of NH$_4^+$ to NO$_3^-$ via nitrite (NO$_2^-$). Denitrification reduces NO$_3^-$ to N$_2$ through the
intermediates NO$_2^-$, NO and N$_2$O. Depending on environmental conditions, this reduction may be
incomplete, leading to N$_2$O emissions. Moreover, the anammox or feammox processes converts of
NH$_4^+$ and NO$_2^-$ to N$_2$ (Ding et al., 2022; Einsiedl et al., 2020). These processes are interconnected
and influenced by environmental conditions, making it challenging to differentiate between the
sources and pathways of N transformations (Nikolenko et al., 2018). Further, functional
characterization of genes encoding key enzymes in N metabolism provides insights into the genetic
potential for specific transformations (Levy-Booth et al., 2014), while microbial community
structure analysis helps elucidate the physiological activities and ecological roles of microbes in
driving N transformations. Together, these approaches contribute to a comprehensive
understanding of N cycling processes.
Stable isotope studies help in tracing the N sources and transformations, through isotopic
signatures such as δ$^{15}$N and δ$^{18}$O in NO$_3^-$, NO$_2^-$, NH$_4^+$, and N$_2$O (Denk et al., 2017, Deb et al.,
2024). However, limitations arise due to overlapping ranges of different isotope sources or
difficulty in distinction between isotope fractionation processes and mixing. To overcome such
limitations and enhance interpretations based on stable isotope studies a multi-compound analysis
approach can be applied (Well et al., 2012, Deb et al., 2024). Such multi-compound isotope
analysis provides a broader perspective on N cycle processes by examining multiple N-compounds
e.g., in denitrification δ$^{15}$N and δ$^{18}$O analysis of NO$_3^-$ helps identify substrates, while NO$_2^-$ and
N$_2$O analyses provide insight into intermediate products.
Since the range of natural isotope variations is relatively narrow, even analyses of multiple
compounds may provide ambiguous results. $^{15}$N tracing technique allows for precise tracing of
artificially added N in the environment (Müller et al., 2004) but is spatially and temporally limited
and disrupts natural abundance isotope studies (Buchen-Tschiskale et al., 2023; Well et al., 2019).,
which are easily and universally applicable in unmodified environmental conditions. A major
drawback of traditional $^{15}$N tracing methods is the necessary sacrifice of other isotope tracers, such
as O isotope signatures and site preference values of N$_2$O, which cannot be accurately determined
when high $^{15}$N additions are applied. This research introduces a novel approach using low-level
$^{15}$N labelling, where a minimal amount of $^{15}$N-labelled substrate is added to slightly increase δ$^{15}$N
values (up to ca. 100-200‰) of a single substrate while maintaining the natural abundance levels.
This ensures that isotope fractionation remains relevant, and standard measurement methods for
all isotope signatures can still be applied. If the level of $^{15}$N labelling exceeds the natural variability
of N sources and isotope fractionation effects, this approach enables clear distinction between
substrates involved in N transformations. It also allows for precise tracing of the path of N from



substrate to product while utilizing or determining isotope fractionation factors, which remain
applicable to natural abundance isotope studies in natural environments.
While stable isotope analysis provides valuable insights into nitrogen pathways, its interpretation
is often complicated by overlapping fractionation effects (Deb et al., 2024). To refine process
identification, microbiological approaches such as quantitative PCR (qPCR) enable the detection
and quantification of key genes involved in nitrogen transformations, providing insights into
microbial activity (Espenberg et al., 2018; Rohe et al., 2020).
However, these microbiological methods only reveal the potential for microbial species to
participate in nitrogen cycling rather than directly quantifying transformation rates. The detection
of functional genes or gene expression does not confirm whether a process is actively occurring at
a given time or its relative contribution within a system (Espenberg et al., 2018; Rohe et al.,
2020)Thus, combining stable isotope data with microbiological analyses enhances the precision of
nitrogen flux assessments in groundwater, offering a robust framework for tracing, quantifying,
and characterizing nitrogen transformations in complex environmental systems. The integration of
isotopic and microbial techniques for partitioning N cycle processes provided valuable insights
into $N_2O$ source apportioning (MASTA et al., 2024).
Here we combine the isotope studies, applying novel low-labelling technique and multicompound
analyses, with microbiological analyses using quantitative PCR (qPCR), which identify and
quantify key genes involved in N processes. This aims at better understanding of the occurring N
transformations and enhancement the precision of nitrogen flux assessments in groundwater.

**2. Materials and Methods**
**2.1. Experimental Site**
Experiments were conducted from the groundwaters collected in an agricultural area near
Wołczyn, Poland, approximately 80 km north of Wrocław. On these crop fields wastewater from
a yeast factory is applied as a natural fertilizer, containing 300 mg $L^{-1}$ of TN (total nitrogen) and
835 mg $L^{-1}$ of TP (total potassium). While this approach supports agricultural production by
reducing reliance on synthetic fertilizers, it is likely to have a significant impact on groundwater
quality, especially by increasing N and P load. Preliminary sampling from piezometers in the study
area conducted on July 2023 revealed nitrate concentrations exceeding 80 mg $L^{-1}$ in the
groundwater, raising concerns about elevated nitrate levels, exceeding the norms for drinking
water of 50 mg $L^{-1}$. The following map (Fig. 1) illustrates the study area of our experiment,
highlighting the locations of piezometers used for groundwater sampling.

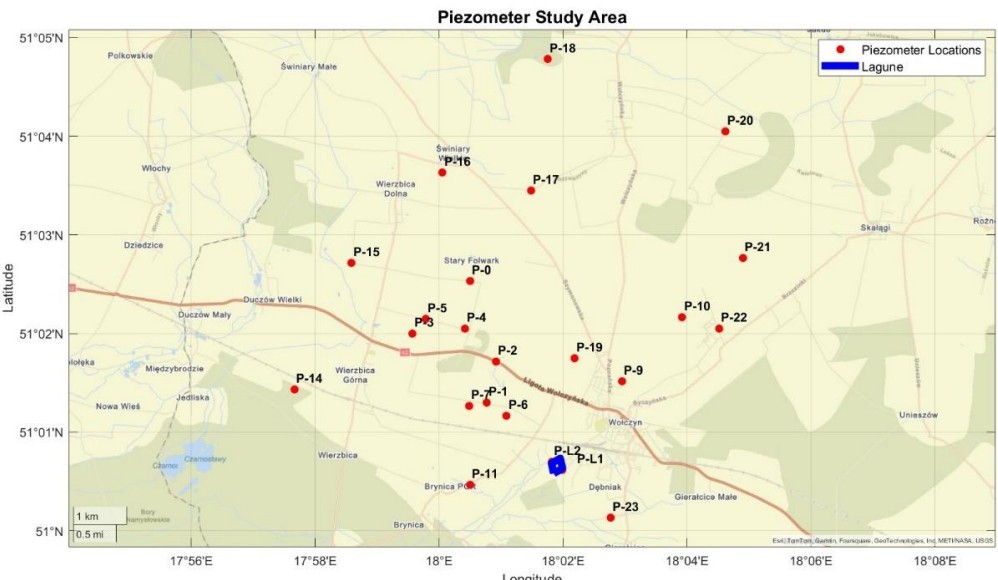

**Figure 1: Piezometer Study Area near Wołczyn, Poland (Blue-marked area indicates the lagune for yeast-production sewage storage).**

The aquifer under study is the top first groundwater horizon connected with surface waters, built of sand-gravel formations of Neogene-Quaternary, hydraulically separated from the underlying Triassic horizon by shale layers. The water table has unconfined character and varying depths from 1.5 to 18.7 m below surface (Olichwer et al., 2012). The thickness of the aquifer range from 4.5 to 31.9 m. The redox potential of the sampled groundwaters varies from 213 to 345 mV and dissolved $O_2$ concentration from 2.2. to 4.3 mg $dm^{-3}$. These values indicate lower $O_2$ content when combined to saturated conditions (ca 10 mg $dm^{-3}$), but slightly higher than typical denitrification favoring conditions (below 2 mg $dm^{-3}$) (Wolters et al., 2022).

## 2.2. Water Sampling

Groundwater samples from 23 piezometers in the study area were pumped out at varying depths during the field sampling campaign on 5th September 2023 (Table 1). Subsequently, water from four selected piezometers (P-7, P-16, P-20, P-23) were used for laboratory incubation studies to evaluate the potential N transformation processes and identify the isotope effects associated with them.



**Table 1: Concentrations of Nitrogen Species and Environmental Parameters in Groundwater Samples** (P abbreviated for piezometer), bd – below detection, piezometers selected for incubation – in bold font.

| Sample | Temperature (°C) | pH | Conductivity (µS cm$^{-1}$) | Nitrate (N-NO$_3^-$) (mg N L$^{-1}$) | Nitrite (N-NO$_2^-$) (mg N L$^{-1}$) | Ammonium (N-NH$_4^+$) (mg N L$^{-1}$) | DOC (dissolved organic carbon) (mg C L$^{-1}$) | DON (dissolved organic nitrogen) (mgN L$^{-1}$) | δ$^{15}$N | | δ$^{18}$O | |
|---|---|---|---|---|---|---|---|---|---|---|---|---|
| | | | | | | | | | δ$^{15}$N-NO$_3^-$ | δ$^{15}$N-NO$_2^-$ | δ$^{18}$O-NO$_3^-$ | δ$^{18}$O-NO$_2^-$ |
| P-0 | 13.5 | 6.5 | 928 | 39.65 | 0.08 | 0.33 | 24.12 | 32.76 | 0.9 | -31.0 | 0.5 | bd |
| P-1 | 14 | 6.1 | 1000 | 21.05 | 0.284 | 0.224 | 42.06 | 16.76 | 7.7 | -27.8 | 6.1 | -7.3 |
| P-2 | 13.4 | 7.3 | 1245 | 0.62 | 0.21 | 7.05 | 67.42 | bd | bd | bd | bd | bd |
| P-3 | 14.1 | 6.7 | 998 | 39.3 | 0.405 | 0.316 | 27.28 | 57.08 | 4.3 | -30.2 | 0.4 | -4.2 |
| P-4 | 12.8 | 7.3 | 4794 | 0.63 | 0.398 | 11.46 | 431.3 | 7.74 | bd | bd | bd | bd |
| P-5 | 13.5 | 6.8 | 1030 | 18.45 | 0.092 | 0.103 | 176.9 | 6.12 | 9.3 | bd | 3.3 | bd |
| P-6 | 17 | 7.5 | 1386 | 0.24 | 0.034 | 0.31 | 58.42 | bd | bd | bd | bd | bd |
| **P-7** | 13.2 | 6.9 | 1395 | 32.8 | 0.188 | 0.138 | 46.32 | 41.04 | 6.6 | -37.0 | 3.2 | 8.5 |
| P-9 | 14.4 | 6.7 | 919 | 0.11 | <0 | 0.17 | 52.59 | bd | bd | bd | bd | bd |
| P-10 | 12.2 | 6.8 | 944 | 0.98 | 0.02 | 1.509 | 24.66 | bd | 10.4 | bd | 6.1 | bd |
| P-11 | 14.2 | 7.3 | 1343 | 0.5 | 0.081 | 2.33 | bd | bd | bd | bd | bd | bd |
| P-14 | 15.5 | 7.1 | 845 | 0.33 | 0.114 | 14.98 | bd | bd | bd | bd | bd | bd |
| P-15 | 14.7 | 7 | 512 | 0.44 | 0.146 | 4.397 | 22.28 | bd | bd | bd | bd | bd |
| **P-16** | 10 | 6.2 | 617 | 39.45 | 0.098 | 0.031 | 10.22 | 31.88 | 3.7 | -17.3 | 1.6 | 4.4 |
| P-17 | 12.7 | 7.1 | 685 | 0.26 | 0.018 | 0.557 | 51.81 | bd | bd | bd | bd | bd |
| P-18 | 11.2 | 7.1 | 560 | 1.8 | 0.06 | 0.651 | 55.85 | bd | 8.4 | bd | 7.7 | bd |
| P-19 | 13.9 | 6.6 | 617 | 0.52 | 0.025 | 0.611 | 9.23 | bd | bd | bd | bd | bd |
| **P-20** | 11.1 | 6.6 | 471 | 38.12 | 0.019 | 0.028 | 9.7 | 30.63 | 1.0 | bd | -0.9 | bd |
| P-21 | 10.9 | 7.1 | 574 | 0.43 | 0.087 | 0.565 | 31.49 | bd | bd | bd | bd | bd |
| P-22 | 11.1 | 5.6 | 557 | 2.47 | 0.064 | 0.215 | 14.81 | 5.01 | 16.0 | bd | 4.8 | bd |
| **P-23** | 14.9 | 6.3 | 1238 | 89.5 | 0.354 | 0.059 | 21.22 | 91.72 | 5.3 | -27.9 | 4.7 | -8.9 |
| P-L1 | 14.2 | 6.8 | 2581 | 1.26 | 0.226 | 0.817 | 262.4 | 69.28 | 10.3 | 5.4 | 1.9 | 15.7 |
| P-L2 | 13.8 | 7 | 1777 | 0.2 | 0.058 | 17.95 | 68.9 | 25.74 | bd | bd | bd | bd |



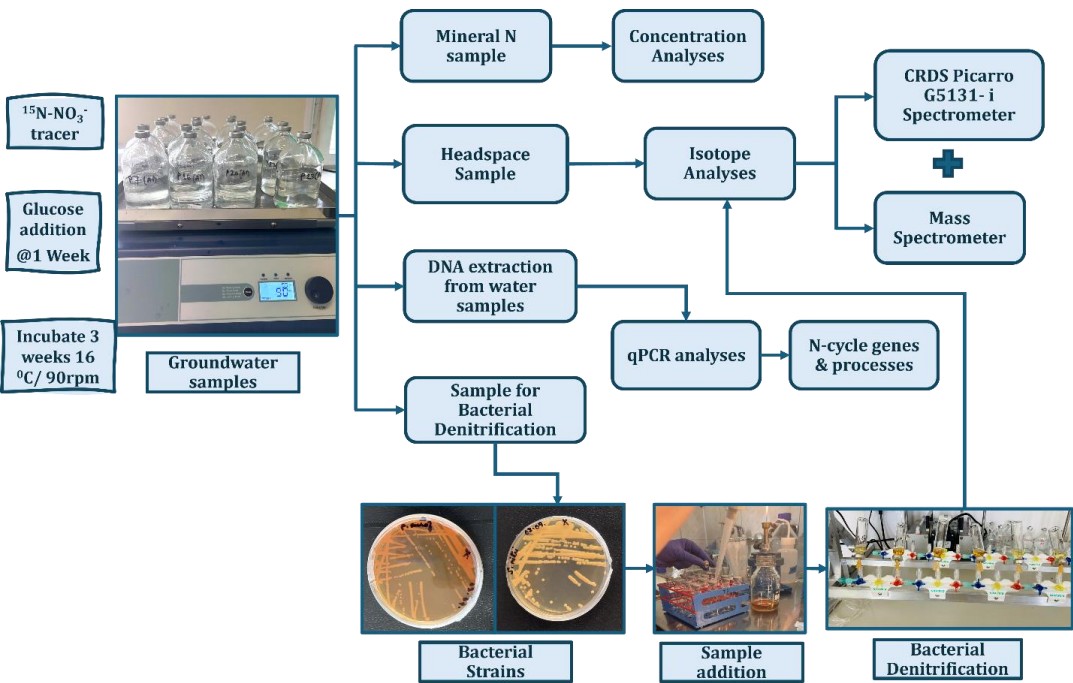

**Figure 2: Experimental Setup for microbial analyses (qPCR, Groundwater Incubation) and Isotopic Analysis.** The scheme illustrates the workflow for analyzing groundwater samples, including incubation with $^{15}N\text{-}NO_3^-$ tracer, bacterial denitrification, DNA extraction, concentration analyses, and isotope measurements using a CRDS (Cavity Ring Down Spectroscopy) Picarro G5131-i spectrometer and mass spectrometer.

For inorganic N concentration and isotopic analyses, all groundwater samples were filtered using 0.45 µm filters. For $NO_3^-$ and $NH_4^+$ analysis, 50 ml of the filtered sample was collected in a Falcon tube, which was stored frozen until further analysis. For $NO_2^-$ analysis, an additional 50 ml of the sample was collected in a separate Falcon tube, where after filtering 1 mL of 2 M KOH was added to raise the pH to 10-12, inhibiting nitrite reduction. The samples were then stored at 4 °C until further analysis. It is essential to analyze these samples as soon as possible after collection to prevent microbial degradation and ensure data integrity.

From the field sampling 4 selected samples with high nitrate concentration were used for filtering and further microbial analyses. The field groundwater samples were immediately transported to the laboratory in an ice-cooled box and filtered using 0.45 µm filters . For laboratory incubation studies, 2 L of groundwater from selected piezometers with high nitrate concentration were collected into sterile bottles and immediately sealed for a series of laboratory experiments, and stored frozen until further analysis. Further, from the later incubation studies (as described in 2.5), the water samples (600 ml from each incubated piezometers) were filtered using sterile 0.45 µm filters for the further microbial analyses after the 3 week incubations period.



**2.3. Inorganic nitrogen analyses ($NO_3^-$, $NO_2^-$, $NH_4^+$) Using a Colorimetric Method**

For the analysis of $NO_3^-$, $NO_2^-$, and $NH_4^+$ concentrations, groundwater samples were filtered using 0.45 μm filters and measured with the SLANDI Photometer LF300 (Slandi Sp. z o.o., Michałowice, Poland), a versatile instrument for water and wastewater analysis across wavelengths ranging from 380 nm to 810 nm. For our analysis, wavelengths of 520 nm, 560 nm, and 610 nm were selected for $NO_3^-$, $NO_2^-$, and $NH_4^+$ concentration, respectively. Following a standardized protocol, specific reagents were added to the samples, allowing the reactions to develop colour and the concentrations were then measured photometrically.

**2.4. Inorganic nitrogen Isotope analyses**

To trace microbial N transformations processes in the groundwater samples, inorganic N isotope analyses were performed with specific bacterial strains *Pseudomonas aureofaciens* for $NO_3^-$ and *Stenotrophomonas nitritireducens* for $NO_2^-$ isotopes. These strains carry out denitrification with $N_2O$ as the end product, as they lack the $N_2O$ reductase gene (Böhlke et al., 2007; Sigman et al., 2001). The detailed laboratory protocol encompassing the preparation and handling of the bacterial species, along with sample addition and isotope analysis is mentioned in previous publication (Deb and Lewicka-Szczebak, 2024). Gas samples were transferred from the headspace to previously evacuated Exetainer vials (Labco Limited, Ceredigion, UK), diluted and analyzed for isotope values $\delta^{15}N$ and $\delta^{18}O$ values of $N_2O$ using mass spectrometry (Thermo Scientific, MAT 253 Plus mass spectrometer combined with GasBench and Precon) in the Laboratory of Isotope Geology and Geoecology at the University of Wrocław, Poland.

**2.5. Laboratory Incubation of Groundwater Samples**

Laboratory incubation studies were conducted with groundwater samples from 4 selected piezometers (P, abbreviated for piezometer: P-7, P-16, P-20, and P-23) with high nitrate concentrations to investigate natural nitrate reduction and identify favorable conditions for denitrification. A volume of 150 mL of groundwater from each piezometer was transferred into sterile 250 mL flasks, with each sample prepared in four replicates along with sterile controls. Sterile samples were produced by filtering the groundwater through 0.45 μm filters followed by the addition of 2 mL of $HgCl_2$ to inhibit microbial activity. These sterile samples served as controls for comparison with active treatments. The incubation flasks with groundwater samples were flushed with $N_2$ gas for 15 minutes, with the flow rate 60–70 mL $min^{-1}$ and 0.6 Bar, to create suboxic conditions of similar $O_2$ content as in the studied aquifer. The final $O_2$ concentration in the headspace was about 5%, which corresponds to the dissolved $O_2$ content in water of 2.1 mg $L^{-1}$ (Table 2). The pH of the samples, approximately 6.5 for each, was maintained without any adjustments. Prior to incubation, a low amount of $^{15}N$-$NO_3^-$ labelled tracer was added to each sample based on its initial nitrate concentration resulting in at% $^{15}N$ (Atom Percent $^{15}N$) of 0.4296%–0.4700%, slightly exceeding the natural abundance (0.366%), to trace N transformation pathways. The target $\delta^{15}N$ value of final $NO_3^-$ was 200‰. A stock solution was prepared by dissolving 12.1429 mg of $Na^{15}NO_3$ (99% $^{15}N$) in 50 mL of water. From this, 1 mL was added to samples P-7, P-16, and P-20, while 2 mL was added to P-23. 1 mL of glucose, equivalent to the addition of 616 mg of C, was added as an additional carbon source after one week of incubation



to stimulate microbial activity and enhance denitrification. All samples were incubated in dark for
three weeks at 16 °C with agitation at 90 rpm.
**Table 2: Data on GC headspace gas analyses for $O_2$, $CO_2$ and $N_2O$.**
The average values and standards deviations of 4 repeated incubation flasks are shown. The
respective: dissolved $O_2$ concentration, $N_2O$ production and $CO_2$ production were calculated
taking into account the gas constant for gases dissolution in water for the incubation temperature
of 16 C. For sterile samples the average data for the samplings before glucose addition (1 and 3)
and after glucose addition (4+6).

| piezometer | sampling | day | $O_2$ [%] | | Dissolved $O_2$ [mg L$^{-1}$]] | $CO_2$ [ppm] | | $N_2O$ [ppb] | | $N_2O$ | $CO_2$ |
|---|---|---|---|---|---|---|---|---|---|---|---|
| | | | average | stdev | | average | stdev | average | stdev | production µg/L/d | production mg/L/d |
| P7 | 1 | 2 | 8.5 | 1.5 | 3.5 | 865 | 225 | 918 | 189 | 0.66 | 1.82 |
| P16 | 1 | 2 | 7.6 | 1.2 | 3.2 | 761 | 184 | 524 | 311 | 0.38 | 1.60 |
| P20 | 1 | 2 | 7.3 | 1.6 | 3.0 | 391 | 48 | 258 | 179 | 0.18 | 0.82 |
| P23 | 1 | 2 | 7.9 | 0.7 | 3.3 | 575 | 50 | 365 | 178 | 0.26 | 1.21 |
| P7 | 3 | 7 | 4.3 | 2.3 | 1.8 | 1755 | 145 | 3316 | 3378 | 0.82 | 2.88 |
| P16 | 3 | 7 | 3.2 | 1.4 | 1.3 | 1129 | 234 | 540 | 561 | 0.13 | 1.85 |
| P20 | 3 | 7 | 3.8 | 1.4 | 1.6 | 337 | 156 | 527 | 547 | 0.13 | 0.55 |
| P23 | 3 | 7 | 4.4 | 1.2 | 1.8 | 762 | 129 | 654 | 257 | 0.16 | 1.25 |
| *flushing + glucose addition* | | | | | | | | | | | |
| P7 | 4 | 9 | 4.2 | 1.5 | 1.7 | 1340 | 294 | 393 | 246 | 0.28 | 2.82 |
| P16 | 4 | 9 | 4.4 | 1.3 | 1.8 | 781 | 329 | 1290 | 1157 | 0.92 | 1.65 |
| P20 | 4 | 9 | 2.6 | 1.1 | 1.1 | 426 | 399 | 133 | 85 | 0.10 | 0.90 |
| P23 | 4 | 9 | 2.4 | 0.9 | 1.0 | 326 | 283 | 497 | 374 | 0.36 | 0.69 |
| P7 | 6 | 14 | 3.9 | 2.2 | 1.6 | 1195 | 1545 | 5993 | 5657 | 1.49 | 1.96 |
| P16 | 6 | 14 | 2.9 | 1.4 | 1.2 | 1823 | 1369 | 46578 | 91114 | 11.58 | 2.99 |
| P20 | 6 | 14 | 2.4 | 1.2 | 1.0 | 1908 | 475 | 4459 | 7284 | 1.11 | 3.13 |
| P23 | 6 | 14 | 4.1 | 2.0 | 1.7 | 1807 | 774 | 2052 | 2050 | 0.51 | 2.96 |
| *Sterile samples* | | | | | | | | | | | |
| P7 | 1+3 | 4 | 4.4 | | 3.0 | 1928 | | 70 | | 0.03 | 3.43 |
| P16 | 1+3 | 4 | 8.1 | | 4.0 | 2160 | | 224 | | 0.09 | 3.84 |
| P20 | 1+3 | 4 | 5.5 | | 2.6 | 632 | | 0 | | 0.00 | 1.12 |
| P23 | 1+3 | 4 | 6.4 | | 3.3 | 1205 | | 0 | | 0.00 | 2.14 |
| | | | | | | | | | | *flushing + glucose addition* | |
| P7 | 4+6 | 4 | 3.5 | | 2.6 | 1381 | | 218 | | 0.08 | 2.46 |
| P16 | 4+6 | 4 | 3.5 | | 1.7 | 1299 | | 1654 | | 0.64 | 2.31 |
| P20 | 4+6 | 4 | 2.3 | | 1.1 | 387 | | 106 | | 0.04 | 0.69 |
| P23 | 4+6 | 4 | 2.6 | | 0.8 | 1044 | | 517 | | 0.20 | 1.86 |






Headspace samples were periodically collected to measure $N_2O$ concentration and isotope
signatures, providing insights into nitrate reduction and denitrification processes under anoxic
conditions. The samples of 25 mL of headspace gas were collected each second day into pre-
evacuated 12 mL Labco Exetainers (1 Bar overpressure). The sampled gas volume was replaced
with pure $N_2$ gas. The headspace samples were analysed on the gas chromatograph Shimadzu GC
Nexis 2030 equipped with barrier discharge ionisation detector (BID) and thermal conductivity
detector (TCD) for $O_2$, $CO_2$ and $N_2O$ concentration (Bucha et al., 2025). The $N_2O$ gas was analysed
for $\delta^{15}N$, $\delta^{18}O$ and $\delta^{15}N^{SP}$ (difference between $\delta^{15}N$ values between central and terminal position
of N in the linear $N_2O$ molecule) using cavity ring-down spectroscopy (CRDS) by Picarro G5131-
i spectrometer equipped with small sample injection module (SSIM) and connected to SRI
autosampler (Eckhardt et al., unpublished) in Laboratory of Isotope Geology and Geoecology at
the University of Wrocław. The isotope analytical limit was about 300 ppb $N_2O$, for this ambient
concentration the measurement precision was better than 0.5‰ for $\delta^{15}N$ and $\delta^{18}O$ and better than
1‰ for $\delta^{15}N^{SP}$. Since this is a newly developed measurement technique, the controlled
measurements for selected sampling points were performed at Thünen Institute, Braunschweig,
Germany applying mass spectrometry (MS) (Thermo Scientific, 5 collector Delta V mass
spectrometer combined with Trace GC and Precon) (Lewicka-Szczebak et al., 2020). After
applying proper corrections to CRDS technique (Harris et al., 2020) and isotope normalization
with the same sets of standards the results between both approaches showed good repeatability
within up to 2‰ difference for $\delta^{15}N$ and $\delta^{18}O$ and up to 4‰ difference for $\delta^{15}N^{SP}$, which fits
withing typical reasonable range for comparing measurements with different techniques (Mohn et
al., 2014). For sterile samples (with $HgCl_2$ addition) the CRDS technique gave erogenous results,
thus only MS results were accepted.
The $N_2O$ isotope results were evaluated using modeling software FRAME (isotope FRactionation
And Mixing Evaluation) (https://malewick.github.io/frame/) to identify $N_2O$ production pathways
and quantify $N_2O$ reduction to $N_2$ (Lewicki et al., 2022).
Inorganic N levels in incubated samples were analyzed at the beginning, after one week of
incubation before glucose addition, and at the end of the experiment. Additionally, the isotopic
signatures of inorganic N were determined using bacterial denitrification by *Pseudomonas*
*aureofaciens* and *Stenotrophomonas nitritireducens*. The $N_2O$ gas formed during this conversion,
representing nitrate or nitrite isotope signatures $\delta^{15}N$ and $\delta^{18}O$, was measured with mass
spectrometry (Thermo Scientific, MAT 253 Plus mass spectrometer combined with GasBench and
Precon).
**2.6. DNA extraction and qPCR analyses for the field and experimental samples**
For DNA extraction, the groundwater samples were filtered using sterile 0.45 μm mixed cellulose
esters (MCE ) membrane filters. The filters were stored at −80 °C for subsequent analysis. DNA
was extracted from 250 mg of water filters using the DNeasy® PowerSoil® Pro Kit (Qiagen,
Germany), following the manufacturer's protocol with a modification: samples were homogenized
using a Precellys 24 homogenizer (Bertin Technologies, France) at 5000 rpm for 20 seconds. DNA



concentration and quality were assessed using a TECAN Infinite M200 spectrophotometer, and
the extracted DNA was stored at −20 °C for further microbial analysis.
qPCR (quantitative Polymerase Chain Reaction) was used to quantify the bacterial and archaeal
16S rRNA genes, as well as the abundances of genes involved in denitrification (*nirS, nirK, nosZI,*
*and nosZII),* nitrification (bacterial, archaeal, and comammox (complete ammonia oxidation)
*amoA*), nitrogen fixation *(nifH) and* dissimilatory nitrate reduction to ammonium (DNRA, *nrfA*).
qPCR reactions were performed using a Rotor-Gene Q thermocycler (Qiagen, Germany). The 10
µl reaction mixture consisted of 1 µl of extracted DNA, forward and reverse gene-specific primers,
5 µl of Maxima SYBR Green Master mix reagent (Thermo Fisher Scientific, Waltham, MA, USA),
and MilliQ water. Each sample was amplified in duplicate, with DNA-free negative control
samples included in every run. The thermal cycling conditions and primers used are detailed in
Table.3 (Espenberg et al., 2024). qPCR results were analyzed using Rotor-Gene® Q software
v.2.0.2 (Qiagen) and LinRegPCR v.2020.2 (Netherlands). The number of gene copies was
calculated based on standard curve ranges (Espenberg et al., 2018; Kuusemets et al., 2024) and
expressed as gene copies per ml of water (copies mL$^{-1}$). DNA extraction and qPCR analysis were
conducted in the Department of Geography at the University of Tartu, Estonia.
**Table 3: qPCR primer pairs and programs for targeted genes**

| Target gene | Primer | Primer concentration (µM) | Program | |
|---|---|---|---|---|
| bacterial 16S rRNA | Bact517F<br>Bact1028R | 0.3 | 95°C 30s; 60°C 45s; 72°C 45s | x 40 cycles |
| archaeal 16S rRNA | Arc519F<br>Arch910R | 0.3 | 95°C 15 s;56°C 30 s; 72°C 30 s | x 45 cycles |
| *nirK* | nirK876<br>nirK1040 | 0.4 | 95°C 15 s, 58°C 30 s; 72°C 30s, 80°C 30 s | x 45 cycles |
| *nirS* | nirSCd3af<br>nirSR3cd | 0.4 | 95°C 15 s, 55°C 30 s; 72°C 30s, 80°C 30 s | x 45 cycles |
| *nosZI* | nosZ2F<br>nosZ2R | 0.4 | 95°C 15 s, 60°C 30 s, 72°C 30 s, 80°C 30s | x 45 cycles |
| *nosZII* | nosZIIF<br>nosZIIR | 1 | 95°C 30 s, 54°C 45 s, 72°C 45 s, 80°C 45 s | x 45 cycles |
| bacterial *amoA* | amoA-1F<br>amoA-2R | 0.4 | 95°C 30 s, 57°C 45 s, 72°C 45 s | x 45 cycles |
| archaeal *amoA* | CrenamoA 23F<br>CrenamoA 616R | 0.4 | 95°C 30 s, 55°C 45 s, 72°C 45 s | x 45 cycles |
| comammox *amoA* | comamoA AF<br>comamoA SR | 0.6 | 95 °C 15 s, 55 °C 30 s, 72 °C 30 s | x 40 cycles |
| *nrfA* | nrfAF2awMOD<br>nrfAR1MOD | 0.6 | 95 °C 15 s, 56 °C 30 s, 72 °C 30 s | x 45 cycles |
| *nifHA* | Ueda19F<br>Ueda407R | 0.4 | 95 °C 30 s, 53 °C 45 s, 72 °C 45 s | x 45 cycles |



## 3. Results
### 3.1. Dissolved inorganic N compounds
#### 3.1.1. Inorganic nitrogen ($NO_3^-$, $NO_2^-$, $NH_4^+$) content and isotope signatures of initial field samples

Initial field samples were measured for inorganic N to determine $NO_3^-$, $NO_2^-$, and $NH_4^+$ concentration and identify piezometers with the highest nitrate levels, which were then selected for laboratory incubation studies. While field measurements provided baseline reference concentrations of $NO_3^-$, $NO_2^-$, and $NH_4^+$, the laboratory incubation samples revealed significant changes in these concentrations over the incubation period, highlighting N transformation processes under controlled conditions.

Initial field samples before the start of incubation showed $NO_3^-$ concentration from 0.2 mg N L$^{-1}$ to 89.5 mg N L$^{-1}$, $NO_2^-$ concentration from 0.02 to 0.4 mg N L$^{-1}$ and $NH_4^+$ concentration from 0.02 to 17.95 mg N L$^{-1}$ (Table 1). The four samples with especially high nitrate concentration level have been selected for further incubation studies (Table 1).

$NO_3^-$ concentration was determined in 23 samples and $NO_2^-$ in 22, with one sample below the detection limit for $NO_2^-$ (Table 1). But out of these 23 samples, isotope analysis of $\delta^{15}$N-$NO_3^-$ and $\delta^{18}O_{NO3}$ was successful on 12 samples, while $\delta^{15}N_{NO2}$ and $\delta^{18}O_{NO2}$ could be analyzed for 7 samples only (Table 1), with the remainder samples below the detection limit for isotopic analysis. $NH_4^+$ isotopic signature was not determined because of very low concentrations below detection limit for the isotope analysis. All the isotope results are presented in the following figures in the frame of literature data for typical nitrate sources and denitrifying processes (Fig. 3A) and typical ammonium sources and nitrifying N transformation processes (Fig. 3B) after (Deb et al., 2024). Such visual presentation is applied for better identification of possible N sources and N transformations.





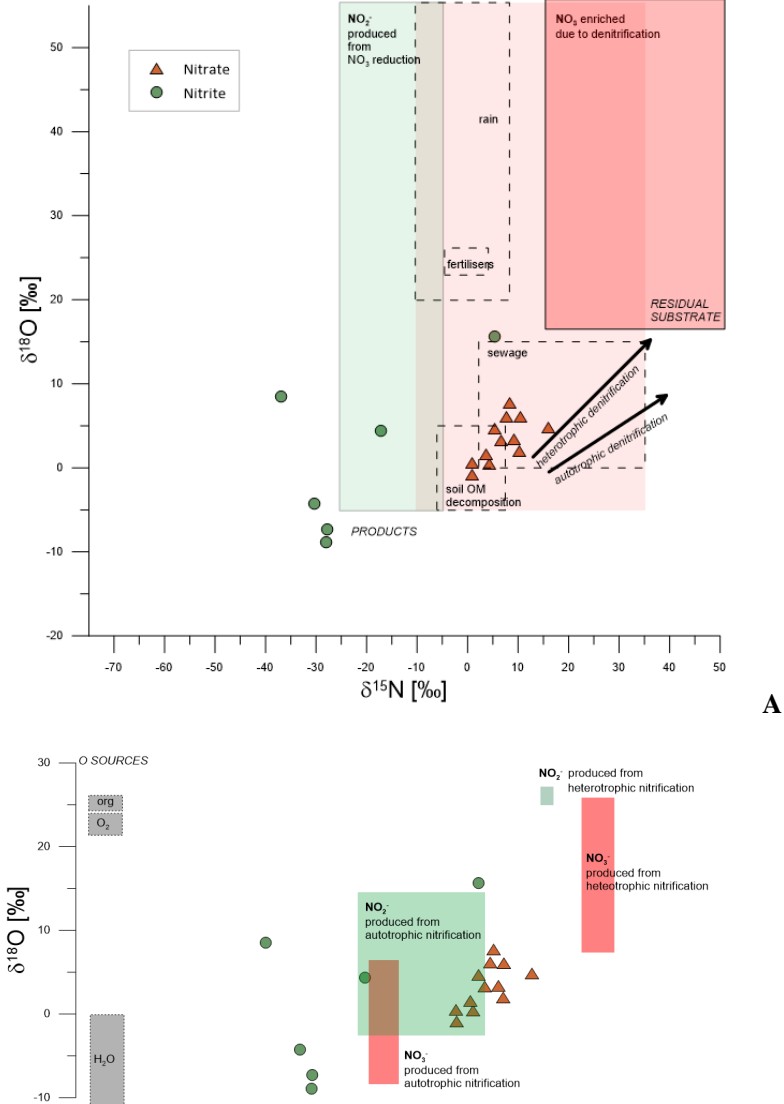

**Figure 3: The isotope signatures of NO₃⁻ (orange triangles) and NO₂⁻ (green circles) in field groundwater samples presented with the literature data for particular N sources and isotope effects for main N transformations, with respect to denitrification processes (A) and nitrification nitrite and nitrate sources (B).** The literature data shown as boxes after (Deb et al., 2024). In (A) NO₃⁻ sources (pink rectangles) include rain, fertilizers, sewage, and soil organic matter, while products (green rectangle) include NO₂⁻, formed during NO₃⁻ reduction. Residual NO₃⁻ enriched through denitrification is represented in the red rectangle. Arrows depict typical



isotope effect associated with autotrophic and heterotrophic processes. In (B) isotopic
characteristics of $NO_2^-$ (green rectangles) and $NO_3^-$ (red rectangles) originating from autotrophic
and heterotrophic nitrification is shown. Grey rectangles illustrate possible oxygen sources ($O_2$ and
$H_2O$) used during nitrifying oxidation processes.
**3.1.2 Inorganic nitrogen ($NO_3^-$, $NO_2^-$, $NH_4^+$) content and isotope signatures during**
**incubation**
During the first phase (before glucose addition), $NO_3^-$ concentrations decreased significantly
across all samples (decrease of 14 to 33 mg $L^{-1}$ N was noted, Fig.4A). Concurrently, $NO_2^-$
concentrations increased significantly reaching around 3.7 to 13.5 mg $L^{-1}$ N, Fig.4A). In the second
phase (after glucose addition), $NO_3^-$ concentrations continue to decrease in all samples (further
decrease of 6.2 to 47.6 mg $L^{-1}$ N compared to day 7 sample, Fig.4A), while $NO_2^-$ levels further
increase for most samples reaching 4.7 to 13.5 mg $L^{-1}$ N. $NH_4^+$ concentrations were very low from
0 to 0.2 mg $L^{-1}$ N) and remained largely unchanged throughout the incubation period.
Further, the isotopic signatures of nitrate ($\delta^{15}N_{NO_3}$ and $\delta^{18}O_{NO_3}$) and nitrite ($\delta^{15}N_{NO_2}$ and $\delta^{18}O_{NO_2}$)
were analysed in water samples during laboratory incubation (Fig 4B). $\delta^{15}N_{NO_3}$ shows much higher
values when compared to initial field samples (Fig. 3) due to low addition of $^{15}N$-$NO_3^-$ tracer. The
preparation of tracer solution and amount of tracer addition was calculated to attain ca. 100-200
‰ as the final $\delta^{15}N_{NO3}$ value. However, due to different initial $NO_3^-$concentrations and precision
of the low amount tracer addition, our final $\delta^{15}N_{NO3}$ after tracer addition is variable for each of the
four incubated samples from approximately 100‰ for P-23 to over 300‰ for P-20 (Fig. 4B).
However, these different final values have been taken into account by all calculations and
modelling, so that the differences did not impact the data interpretation. All calculations were
applied individually for each incubated water sample and individual $\delta^{15}N_{NO3}$ values have been
accepted as the incubation starting point for each of the four water samples. $\delta^{15}N_{NO_3}$ and $\delta^{18}O_{NO_3}$
increase significantly during the first phase of incubation and remain quite stable during the second
incubation phase across all samples. $\delta^{15}N_{NO_2}$ shows slight variability across all samples, with values
ranging from approximately -50‰ to 0‰, hence much lower than the respective $\delta^{15}N_{NO_3}$ values.
$\delta^{18}O_{NO_2}$ shows very dynamic variations without very consistent trends, reflecting the complexity
of microbial and environmental interactions affecting nitrite transformation. Interestingly, there is
very clear pattern for P-7, P-16 and P-20 with significant $\delta^{18}O_{NO_2}$ enrichment for the 2$^{nd}$ sampling
(7 days) and further depletion for the 3$^{rd}$ sampling (14 days) (Fig. 4B).
In the sterile treatment the $NO_3^-$ reduction is even faster than in other samples, while the isotope
signatures are very stable showing very minor isotope enrichment. $NO_2^-$ concentrations are very
low not exceeding 0.3 mg N $L^{-1}$. $NH_4^+$ concentrations increase during the incubation reaching up
to 4 mg N $L^{-1}$, which is much higher than observed for non-sterile samples.





**Figure: 4. Content of inorganic nitrogen forms (orange line: NO₃⁻, green line: NO₂⁻, grey line: NH₄⁺) and their isotopic signatures (δ¹⁵N and δ¹⁸O) during laboratory incubation.** The graphs in A show concentrations variation in time and graphs in B depict changes in δ¹⁵N (red



lines) and $\delta^{18}O$ (blue lines) values over time for nitrate (solid line) and nitrite (dashed line) in
different samples (P, abbreviated for piezometer: P-7, P-16, P-20, and P-23), illustrating dynamic
isotopic variations influenced by microbial processes. Sterile samples are shown as the individual
points on the graphs (for $NO_3^-$ and $NH_4^+$ contents, while $NO_2^-$ was very low for all the sterile
samples, and is not shown).
**3.2. Gas headspace sample analysis**
For almost all samplings significant $N_2O$ and $CO_2$ fluxes were observed during the incubation. The
gases were accumulated in the headspace until day 7 of the incubation (phase 1: day 1 – day 7),
then after flushing the accumulation was started again for next 7 days (phase 2: day 8 – day 14).
Table 2 shows results of headspace gas analyses for the second and last day of the accumulation,
for each incubation phase, before and after glucose addition. The $CO_2$ production varies from 0.7
to 3.1 mg $L^{-1}$ $day^{-1}$ with similar flux range for both phases. The $N_2O$ production varies from 0.1 to
11.6 μg $L^{-1}$ $day^{-1}$ with significantly higher fluxes for the second incubation phase, with one
extremely high outlier.
Sterile samples show $CO_2$ production in the comparable amount to unsterile treatments, and much
lower $N_2O$ production, however still significant for some points, especially for the second
incubation phase after glucose addition (Table 2).
Figure 5 shows the isotopic signatures ($\delta^{15}N^{SP}$, $\delta^{18}O$, $\delta^{15}N$) of $N_2O$ in headspace samples from
laboratory incubation before and after glucose addition together with the main $N_2O$ production
pathways and typical $N_2O$ reduction line summarized after literature data (Yu et al., 2020). The
isotope characteristics for the main $N_2O$ production pathways: bacterial and fungal denitrification
(bD and fD), nitrifier denitrification (nD), nitrification (Ni) and chemodenitrification (chD) are
shown for the particular substrate isotopic signatures of the actual case study: $\delta^{18}O_{H2O}$ of -9.0‰
(mean common value for all water samples) and respective $\delta^{15}N_{NO3}$, separately for each sampling
point (respective values in Table 1).










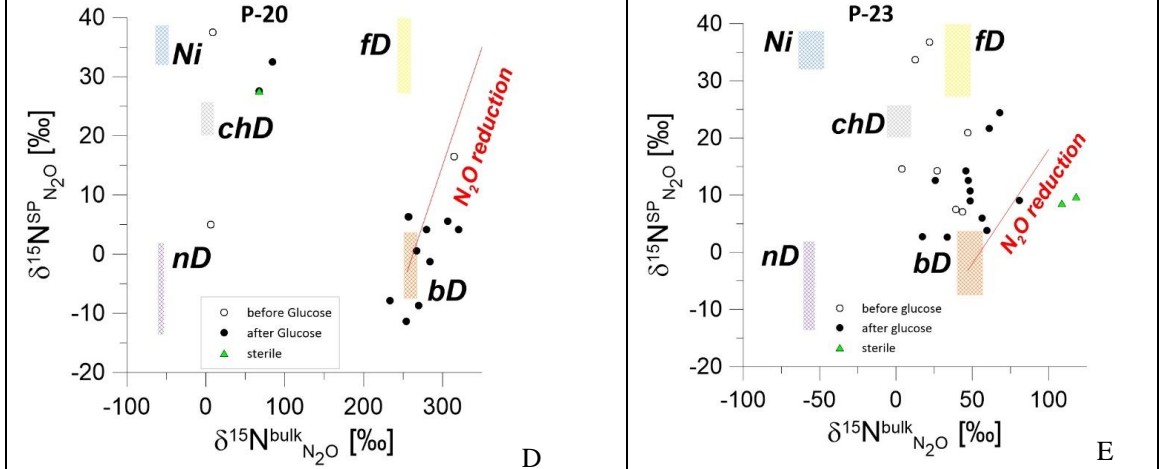

**Figure 5 : Isotopic signatures ($\delta^{15}N^{SP}_{N2O}$, $\delta^{18}O_{N2O}$ and $\delta^{15}N_{N2O}$) highlighting N₂O dynamics**
**and microbial nitrogen transformation pathways during laboratory incubation for**
**groundwater samples (P, abbreviated for piezometer: P-7, P-16, P-20, and P-23).** Empty
circles represent the first incubation phase, filled circles – the second incubation phase after
glucose addition and green triangles show sterile samples. Clustering reflects a shift from mixed
nitrification and denitrification before glucose addition to bacterial denitrification dominance after
glucose addition. Panel A presents $\delta^{15}N^{SP}$-$\delta^{18}O$ map for all samples, since the source processes are
common for all samples, panels B-E present $\delta^{15}N^{SP}$-$\delta^{15}N$ maps individually plotted for each
piezometers, because depending on the particular $^{15}N$ content for each piezometer the mixing
endmembers isotopic signatures (bD and fD) differ. Each plot shows isotopic values before
glucose addition (white circles) and after glucose addition (black circles), reflecting microbial
processes like bacterial denitrification (bD), autotrophic nitrification (Ni), nitrifier denitrification
(nD), and fungal denitrification (fD), with N₂O reduction along the red line.
**3.3 Gene abundance and proportion analyses**
The gene abundance graph (Fig 6A) illustrates the quantification of key nitrogen cycle genes while
the proportions of functional genes relative to total prokaryotic abundance are shown in Fig 6B for
groundwater samples (P-7, P-16, P-20, and P-23) before and after incubation.

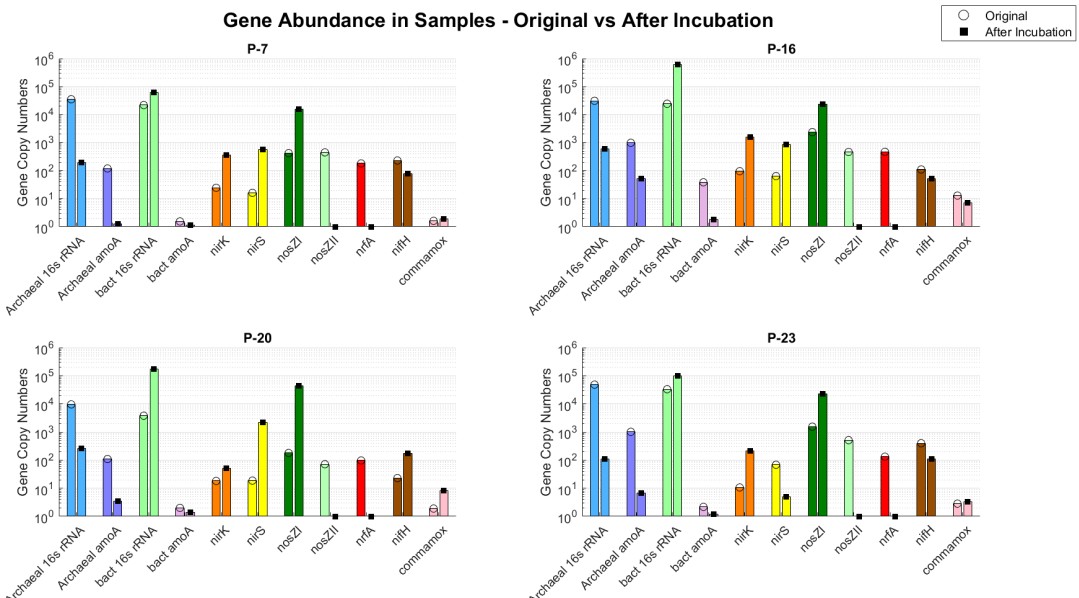

Figure 6 (A): Comparison of gene abundance in groundwater samples before and after incubation

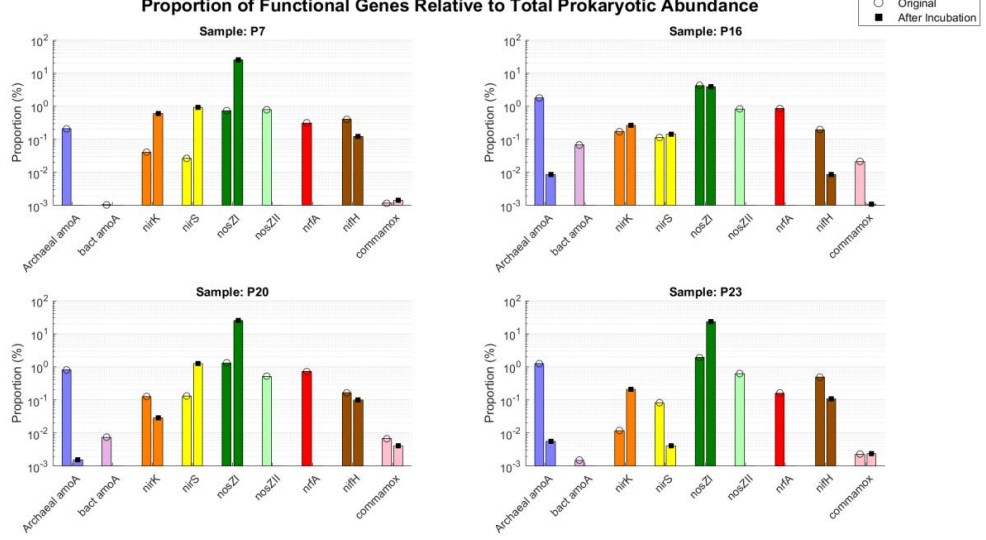

Figure 6 (B): Functional Gene Proportions Pre- and Post-Incubation

The graphs (Fig. 6A and Fig. 6B) illustrate the abundance and proportions of key nitrogen cycle genes in groundwater samples (P-7, P-16, P-20, and P-23) before and after incubation. Figure 6A shows the relative abundance of genes involved in nitrification (archaeal *amoA*, bacterial *amoA*), denitrification (*nirK, nirS, nosZI, nosZII*), nitrogen fixation (*nifH*), and DNRA (*nrfA*), as well as complete nitrification (*commamox*) alongside microbial population markers (archaeal and



bacterial 16s rRNA). Figure 6B presents the proportions of these functional genes relative to total
prokaryotic abundance, highlighting their contributions to the microbial community structure.
**4 Discussion**
**4.1 Initial groundwater samples – N transformations occurring in field conditions**
For identification of possible N-transformations occurring naturally in the aquifer the isotope
signatures of inorganic N ($NO_3^-$ and $NO_2^-$) were compared with the literature data on
denitrification (Fig. 3A) and nitrification (Fig. 3B).
Figure. 3A illustrates the isotopic composition ($\delta^{15}N$ and $\delta^{18}O$) of $NO_3^-$ in groundwater samples,
highlighting both the sources of nitrate and the processes that transformed it during the residence
time in the aquifer. Samples in the pink-shaded zone reflect $NO_3^-$ contributions from sources such
as fertilizers, sewage, and soil organic matter decomposition, with minimal microbial
transformation. Samples with enriched $\delta^{15}N$ and $\delta^{18}O$ values, located in the red-shaded "Residual
Substrate" zone, indicate advanced denitrification, where residual nitrate is enriched in $^{15}N$ and
$^{18}O$ due to preferential reduction of light isotopes. The green-shaded area represents $NO_2^-$
produced from $NO_3^-$ reduction during partial denitrification, an intermediate step in denitrification.
The isotopic patterns in the graph also differentiate between autotrophic and heterotrophic
denitrification. Samples aligning with autotrophic denitrification indicate the reduction of $NO_3^-$
coupled with the oxidation of inorganic compounds like sulfur or hydrogen, resulting in a slower
rate of isotopic fractionation and less pronounced enrichment in $\delta^{15}N$ and $\delta^{18}O$ (Cui et al., 2019;
Hu et al., 2024). In contrast, samples reflecting heterotrophic denitrification show rapid isotopic
fractionation due to the use of organic carbon as the electron donor, leading to greater enrichment
of $\delta^{15}N$ and $\delta^{18}O$ in the residual nitrate (after (Deb et al., 2024)).
Our $NO_3^-$ samples (orange triangles, Fig. 3A) are located in the area typical for $NO_3^-$ originating
from organic matter decomposition and of sewage origin. The precise knowledge of the $\delta^{15}N$
signature of the potential N substrates, i.e. of DON and waste waters, could further confirm the
dominant source of the samples (Boumaiza et al., 2024). The nitrate samples present a clear
correlation between $\delta^{18}O$ and $\delta^{15}N$ values, typical for isotope enrichment due to heterotrophic
denitrification leading to $^{18}O$ and $^{15}N$ enriched of the residual $NO_3^-$. However, this enrichment is
relatively low and the samples do not show typically high $\delta$ values (Fig. 3A). This indicates that
the nitrate pool might be constantly renewed with fresh substrate of low $\delta$ values.
Our $NO_2^-$ samples (green circles, Fig. 3A), are mostly shifted towards lower $\delta^{15}N$ values, with the
expected isotope effect typical for denitrification $NO_3^-$ reduction to $NO_2^-$, with isotopically
depleted $NO_2^-$ due to the preferential reduction of light isotopes ($^{14}N$ and $^{16}O$). $\delta^{18}O_{NO2}$ values are
similar or lower than the respective $NO_3^-$ source, which may indicate additional incorporation of
water into the formed $NO_2^-$.
Figure .3B illustrates the isotopic composition of nitrate ($NO_3^-$) and nitrite ($NO_2^-$) in groundwater
samples in regard to possible nitrification processes. Autotrophic nitrification, with $NO_3^-$ produced
from $NH_4^+$ or organic nitrogen, is characterized by lower $\delta^{15}N$ and $\delta^{18}O$ values, while heterotrophic
nitrification contributes to $NO_3^-$ and $NO_2^-$ production with distinct isotopic enrichment from
organic nitrogen compounds. Our $NO_3^-$ samples are located between values typical for $NO_3^-$





production from autotrophic and heterotrophic nitrification, the observed correlation might be a
mixing between these two $NO_3^-$ origins. However, from the $NO_2^-$ samples only two points indicate
typical values for autotrophic nitrification, whereas others show much lower $\delta^{15}N$ values (Fig. 3B).
Both graphs show the potentially occurring processes, it is important to review them jointly with
the basic aquifer information and further microbial analyses and incubation studies. The
physicochemical parameters for our aquifer present redox conditions theoretically allowing for
occurrence of both denitrification and nitrification processes. Oxygen concentrations in the range
of less than 1 and up to 2 mg $O_2$ $L^{-1}$ are regarded as the boundary between nitrate-reducing and
non-nitrate-reducing conditions in groundwater (Wolters et al., 2022). Hence, the range of
dissolved oxygen content observed for the aquifer under study of 2.2 - 4.3 mg $O_2$ $L^{-1}$ is slightly
higher, and denitrifying processes might be suppressed. The redox potential of our aquifer of 213-
345 mV lies also on the edge of typical denitrifying conditions from 10 to 300 mV (Brettar et al.,
2002). This suggests that reduction processes might occur but might be also accompanied by
oxidation processes. Consequently, both conclusions drawn from the Fig. 3A and 3B might be
simultaneously true. Whereas $NO_3^-$ is being denitrified it might be simultaneously produced both
in autotrophic and heterotrophic nitrification, which is supported by only small $NO_3^-$ enrichment.
Groundwater samples of dominant denitrification show much higher $NO_3^-$ isotope signatures
(Clague et al., 2019). Similarly, $NO_2^-$ isotopic signature shows most probably a mixture of $NO_3^-$
reduction and formation due to nitrification, in various proportions for different samples. There is
one sample of the highest $\delta^{18}O_{NO2}$ and $\delta^{15}N_{NO2}$ (Fig. 3B). This is the P-L2-1 piezometer located
closest to the lagune of yeast sewage storage, the sample of the highest $NH_4^+$ content (Table 1). In
this sample $NO_2^-$ must originate mostly from autotrophic nitrification from ammonium oxidation,
as it can be concluded from Fig. 3B.
The conclusion of active nitrification processes is reinforced with the gene abundances observed
in field samples, before incubation, Fig. 6, where the majority of gene copy numbers represent
archaeal *amoA*, while denitrification genes occurrence is very low. Hence, we rather have intensive
nitrate production by nitrification processes (Fig. 3B).
To further figure out which potential N transformations are active in the aquifer we performed the
laboratory incubations in controlled conditions of selected water samples.
**4.2 Active N transformation processes during incubation**
**4.2.1. Inorganic N analyses**
The dynamic variations in inorganic N concentration and isotopic evolution of $NO_3^-$ and $NO_2^-$
during the laboratory incubation experiments (Fig. 4) across all incubated samples (P-7, P-16, P-
20, and P-23) reflects active microbial transformations during the incubation period. Prior to
glucose addition, the observed decrease in $NO_3^-$ concentration, coupled with a parallel increase in
$\delta^{18}O_{NO3}$ and $\delta^{15}N_{NO3}$ (Fig.4), suggests intensive denitrification with preferential reduction of light
isotopes resulting in enrichment of the residual nitrate. The apparent isotope effect, i.e. the
difference between the initial and final (after 7 days) $NO_3^-$ isotope signature is from 20 to 33‰ for
$\delta^{15}N_{NO3}$ and from 12 to 18‰ for $\delta^{15}N_{NO3}$ giving O/N ratio from 0.45 to 0.83, which is typical slope
for heterotrophic denitrification (from 0.48 to 0.88) (Boumaiza et al., 2024; Clague et al., 2019).





During this first phase the $NO_2^-$ concentration clearly increase from near 0 to a few mg $NO_2^-$ $L^{-1}$
and $\delta^{15}N_{NO_2}$ values show slight increase (Fig.4). This shows that the elevated $\delta^{15}N_{NO3}$ (due to low-
level labeling) is partially transferred to the $NO_2^-$ pool. However, the low magnitude of this
increase is rather surprising, i.e., the $\delta^{15}N_{NO_2}$ do not approach the high values of $NO_3^-$, but increase
only slightly. This indicates that the formed nitrite must partially originate from another $^{15}N$-
depleted pool (unlabelled). Based on the observed $\delta^{15}N_{NO3\_0}$ (initial value, day 0) and change in
$\delta^{15}N_{NO_2}$ values (between day 0 and day 7 of the incubation) we can calculate the maximal
contribution of $NO_2^-$ originating from $NO_3^-$ reduction (NAR) in this new-formed $NO_2^-$ applying the
isotope mass balance (Eq.1). These calculations are simplified by neglecting any isotope
fractionation of the $NO_2^-$ pool.
$$NAR = \frac{\delta^{15}N_{NO2-7} - \delta^{15}N_{NO2-0}}{\delta^{15}N_{NO3-0}}$$    (Eq.1)


**Table 4: Inputs and results of the mass balance calculations for determining the contribution**
**of nitrate reduction (NAR) in the nitrite pool** with Eq.1: $\Delta\delta^{15}N_{NO_2}$ is the change of nitrite N
isotope signature between day 0 and day 7, $\delta^{15}N_{NO3-0}$ - initial N isotope signature of nitrate, $\Delta$
$[NO_2^-]$ $_2$ is the change of nitrite concentration between day 0 and day 7, mg/L NAR is the amount
of produced nitrite originating from NAR, $\Delta$ $[NO_3^-]$ is the nitrate consumption between day 0 and
day 7.

| Piezometer | $\Delta\delta^{15}N_{NO_2}$ | $\delta^{15}N_{NO3-0}$ | NAR [%] | $\Delta$ [NO₂-] mg N L⁻¹ | mg/L NAR | $\Delta$ [NO₃-] mg N L⁻¹ |
|---|---|---|---|---|---|---|
| P-7 | 33.1 | 192.2 | 17.2 | 7.3 | 1.3 | -13.7 |
| P-16 | 18.1 | 207.1 | 8.7 | 13.4 | 1.2 | -23.0 |
| P-20 | 13.4 | 306.5 | 4.4 | 3.6 | 0.2 | -17.2 |
| P-23 | 14.1 | 69.0 | 20.4 | 6.1 | 1.2 | -33.2 |


The calculation results (Table 4) indicate that from 0.2 up to 1.3 mg $N-NO_2^-$ $L^{-1}$ originates from
NAR, which is low amount when compared to the magnitude of $N-NO_3^-$ consumption of 13.7 to
33.2 mg $L^{-1}$ (Table 4). This agrees with the fact that $NO_2^-$ is a very reactive and short living
compound and as denitrification intermediate it instantaneously undergo further reduction
(Lewicka-Szczebak et al., 2021). But interestingly, the large majority of $N-NO_2^-$ (80 to 95 %)
originates from other transformations than $NO_3^-$ reduction. Since the $NH_4^+$ contents are very low
in all the samples, this unlabelled N source for $NO_2^-$ production must originate from dissolved
organic N (DON). This pathway is very plausible since the samples show high DON contents from
31 to 92 mg N $L^{-1}$ (Table 2). For most samples, $NO_2^-$ show significant increase in $\delta^{18}O_{NO_2}$ values
in the first phase (between day 0 and day 7), indicating that the major source of O must be
molecular $O_2$ with characteristic high $\delta^{18}O_{O2}$ of +23.5‰ (Moore et al., 2006). Since the incubations
applied suboxic atmosphere (up to 5% in the headspace and 2.1 mg of dissolved oxygen (Table 2),
this low amounts of oxygen must have been used or the oxygen must had been fixed before in
other compounds, like organic matter, and further used for oxidation processes. Only for P-23 the
$\delta^{18}O_{NO_2}$ value stays stable, this sample shows most intensive $NO_3^-$ reduction due to denitrification





and most probably the potential increase was masked with O-atoms exchange between water and
denitrification intermediates (Lewicka-Szczebak et al., 2016).
In the second phase of the incubation, after glucose addition, further $NO_3^-$ reduction was observed
in all samples (Fig. 4A). However, despite this observed reduction, δ value stay quite stable, with
much less isotope enrichment between day 7 and 14 of the incubation, when compared to the day
0 - day 7 enrichment, both for $\delta^{15}N$ and $\delta^{18}O$ (Fig. 4B). Hence, we do not observe here the typical
isotope enrichment characteristic for denitrification processes (Boumaiza et al., 2024). The
occurrence of intensive denitrification during the second incubation phase can be clearly proved
with $N_2O$ data, which show high $^{15}N$ content, and $\delta^{15}N^{SP}$ and $\delta^{18}O$ values typical for bacterial
denitrification (Fig. 5). Also analysed gene abundances clearly indicate intensification of
denitrification genes during the incubation (Fig. 6). But despite active denitrification process, the
typical isotope enrichment is not observed. This might possibly indicate significant additional
contribution of other process of nitrate reduction. Chemodenitrification can be considered, since
this process is associated with no kinetic isotope effects for either $\delta^{15}N$ or $\delta^{18}O$ in the residual
$NO_3^-$ pool (X. Wang et al., 2022). This assumption can be reinforced with the sterile samples data,
where nitrate pool is also largely reduced (Fig. 4A) without any isotope effects (Fig. 4B). This
indicates that the conditions in the studied groundwaters support chemodenitrification.
Simultaneously, $\delta^{15}N_{NO_2}$ mostly go down or increase only slightly, indicating that the
transformations of unlabelled N source are getting even more active than in the first incubation
phase and there is nearly no detectable contribution of $NO_2^-$ from $NO_3^-$ reduction. However, the
labelled $^{15}N$ is present in the further denitrification product – $N_2O$, hence it must have been
transformed through $NO_2^-$ as the first denitrification intermediate. This shows that this conversion
takes place very rapidly, maybe even in the same microbial cell and $NO_2^-$ must be nearly
completely converted to further denitrification products. Importantly, the common pool of $NO_2^-$,
which do not show $^{15}N$ enrichment, is mostly not converted to $N_2O$. This is proven by the fact that
$\delta^{15}N_{N2O}$ values are very close to $\delta^{15}N_{NO3}$, but much higher than $\delta^{15}N_{NO2}$ during the second
incubation phase. Hence, the $NO_2^-$ newly formed in nitrification processes is not further reduced
to $N_2O$ but is most probably rather further oxidised to $NO_3$. Since this process would add $^{15}N$
depleted $NO_3^-$ this can mask the $^{15}N$ enrichment due to denitrification. In this second incubation
phase, O isotope signatures of $NO_2^-$ and $NO_3^-$ mostly move towards each other, which indicates
probably intensive reversible reactions of reduction and oxidation between these two compounds,
which facilitates O-atoms exchange with water. This agrees with the recent findings by (Zheng et
al., 2023) who indicated tighter cycling between these both compounds with particular importance
of $NO_2^-$ re-oxidation processes. The inconsistencies found in our data for $^{15}N$ content in $NO_3^-$,
$NO_2^-$ and $N_2O$ pool reinforces the assumption of separate $NO_2^-$ pools for particular N
transformation pathways (Müller et al., 2014; Rütting and Müller, 2008; Zhang et al., 2023).
Although most of these previous studies apply for soils, it is apparently also true for groundwater
N transformations.
A closer look on the isotopic analysis of nitrite ($NO_2^-$) in the groundwater incubation study (Fig.
7) reveals that the highest nitrite concentrations are characterised by the lowest isotope signatures.



We observe a statistically significant correlation between O and N isotopic signatures of $NO_2^-$.
Theoretically, this could be the mixing line with the [15]N labelled values originating from labelled
$NO_3^-$, reduction, but this $NO_3^-$, shows low $\delta^{18}O$ values in the range from -16 to +10‰. Hence, this
rather shows mixing of the different origins of unlabelled $NO_2$, which is in great majority (as
shown above and in Table 4), potentially originating from both autotrophic and heterotrophic
nitrification processes.

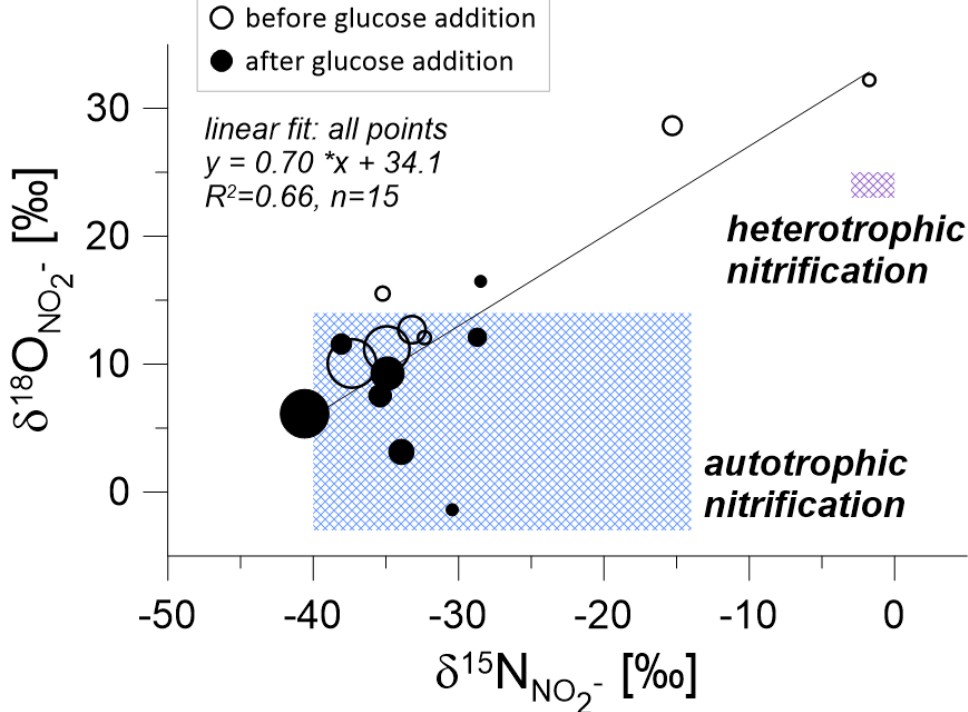

**Figure 7: Isotopic Signatures of Nitrite ($NO_2^-$) during laboratory Incubation: first phase,**
**before glucose addition: empty circles, second phase, after glucose addition: filled circles.**
The points size is proportional to nitrite concentration. The shaded regions correspond to isotopic
ranges associated with autotrophic and heterotrophic nitrification (after (Deb et al., 2024)),
illustrating a shift in processes following glucose addition.
Before the addition of glucose, the isotopic signatures (empty circles) predominantly cluster within
the autotrophic nitrification zone, indicating that autotrophic bacteria initially dominated nitrite
production by converting ammonia to nitrite using $CO_2$. Autotrophic bacteria, such as
*Nitrosomonas europaea* convert ammonia ($NH_3$) to nitrite ($NO_2^-$) as an energy-generating process
(Deb et al., 2024). During this process, $CO_2$ serves as the sole carbon source for these bacteria,
assimilated into their biomass to support cellular growth, independent of the chemical reaction
used for energy generation (Hommes et al., 2003). In the groundwater samples, $CO_2$ likely





originated from the decomposition of organic matter in the yeast sewage (Section 2.1) or from the
carbonate system naturally present in groundwater (Section 3.1.1).
The application of yeast-based sewage as fertilizer in the agricultural site likely introduced organic
nitrogen into the groundwater, which undergoes microbial decomposition to release ammonium
($NH_4^+$) through the mineralization of proteins and amino acids (Watanabe et al., 2023). However,
$NH_4^+$ was not detected in the groundwater samples ( Table 1), indicating its rapid transformation
within the nitrogen cycle. The absence of $NH_4^+$ can be attributed to its immediate consumption
through autotrophic nitrification, where ammonia is oxidized to $NO_2^-$ and $NO_3^-$. The clustering of
isotopic data within the autotrophic nitrification zone in Fig. 7 before glucose addition supports
this process, highlighting ammonia oxidation as a dominant pathway. Additionally, microbial
assimilation likely contributed to $NH_4^+$ exhaustion, as microbes utilized it for biomass synthesis.
This is not the case for sterile samples, where we observe slight accumulation of $NH_4^+$.
Following glucose addition, the isotopic signatures (filled circles) become more dispersed, with
several data points shifting towards the heterotrophic nitrification zone. This shift suggests that
glucose addition stimulated heterotrophic bacteria, altering nitrite production pathways. While
autotrophic nitrification likely continued, the spread in isotopic values indicates that heterotrophic
nitrifiers also contributed to nitrite production, likely utilizing organic nitrogen compounds,
supporting a transition from predominantly autotrophic to mixed nitrification processes. Together,
these findings demonstrate the rapid transformation of $NH_4^+$ from yeast-based fertilizers into
intermediate nitrogen compounds, driving nitrification and subsequent nitrogen cycling processes
in groundwater.

### 642 4.2.2 Headspace N₂O analyses and FRAME isotope model

Figure 5 illustrates the isotopic signatures ($\delta^{15}N^{SP}_{N2O}$, $\delta^{18}O_{N2O}$ and $\delta^{15}N_{N2O}$) of N₂O in headspace
samples collected during laboratory incubation with the relevant microbial N transformation
pathways presented based on the literature data (Yu et al., 2020) and taking into account the actual
measured isotopic signatures of sources applied in this case study ($\delta^{18}O_{H2O}$, $\delta^{15}N_{NO3}$). Before
glucose addition (white dots), the isotopic signatures indicate mixing between nitrification and
denitrification processes (Fig. 5A). In P-7, before glucose addition, isotopic data clustered near the
nitrifier denitrification (nD) zone, highlighting ammonia oxidation and partial N₂O reduction. In
P-16, isotope signatures widely distributed between nitrifier and bacterial denitrification zones,
suggesting overlapping processes. In P-20 and P-23, clustering near the bacterial denitrification
(bD) zone reflected nitrate reduction as the dominant pathway with minimal N₂O reduction.
After glucose addition, the isotopic data indicate that N₂O production was primarily driven by
bacterial denitrification (bD), with relatively low $\delta^{15}N^{SP}_{N2O}$ and $\delta^{18}O_{N2O}$ values, clustering mostly
around reduction line. In the $\delta^{15}N^{SP}$- $\delta^{15}N$ space, mostly the isotopic data showed a clear shift
toward bD, supported by a significant increase in $\delta^{15}N_{N2O}$ values, indicating N₂O production from
the slightly $^{15}N$-labeled $NO_3^-$ pool and some effect of N₂O reduction. In P-23, however, the data
indicate more possible pathways mixture, including nitrification (Ni) and fungal denitrification
(fD). Significant reduction of N₂O to N₂ can be supposed based on the clustering of the points
along the N₂O reduction line, especially in the $\delta^{15}N^{SP}$- $\delta^{18}O$ isotope map, Fig. 5). In $\delta^{15}N^{SP}$- $\delta^{15}N$





isotope map the effect of N$_2$O reduction is less visible, because the artificially elevated $\delta^{15}$N values
result in very steep reduction line. Minimal clustering near the fungal denitrification (fD) zone
suggests limited fungal contributions to N$_2$O production. However, the $\delta^{15}$N$^{SP}$- $\delta^{18}$O map shows
some samples near the fD zone (Fig. 5A), and the FRAME model (Fig. 8) also supports this with
minor yet detectable fungal involvement. The shift in $\delta^{15}$N$^{SP}_{N2O}$ values also reflects the changing
dynamics, with nitrification and nitrifier denitrification becoming less prominent as bacterial
denitrification intensified. In conclusion, the isotopic data demonstrate that carbon availability
strongly influenced the balance between N$_2$O production and reduction, driving microbial N
transformations and regulating N$_2$O emissions during laboratory incubation.
These isotope results of N$_2$O from the headspace samples were jointly analyzed using the three
dimensional FRAME (FRActionation and Mixing Evaluation) model (Lewicki et al., 2022) to
quantitatively interpret the isotopic signatures of N$_2$O, identifying microbial pathways driving N$_2$O
production and estimating N$_2$O reduction progress. This offers most precise insight into N
transformations under controlled experimental conditions.

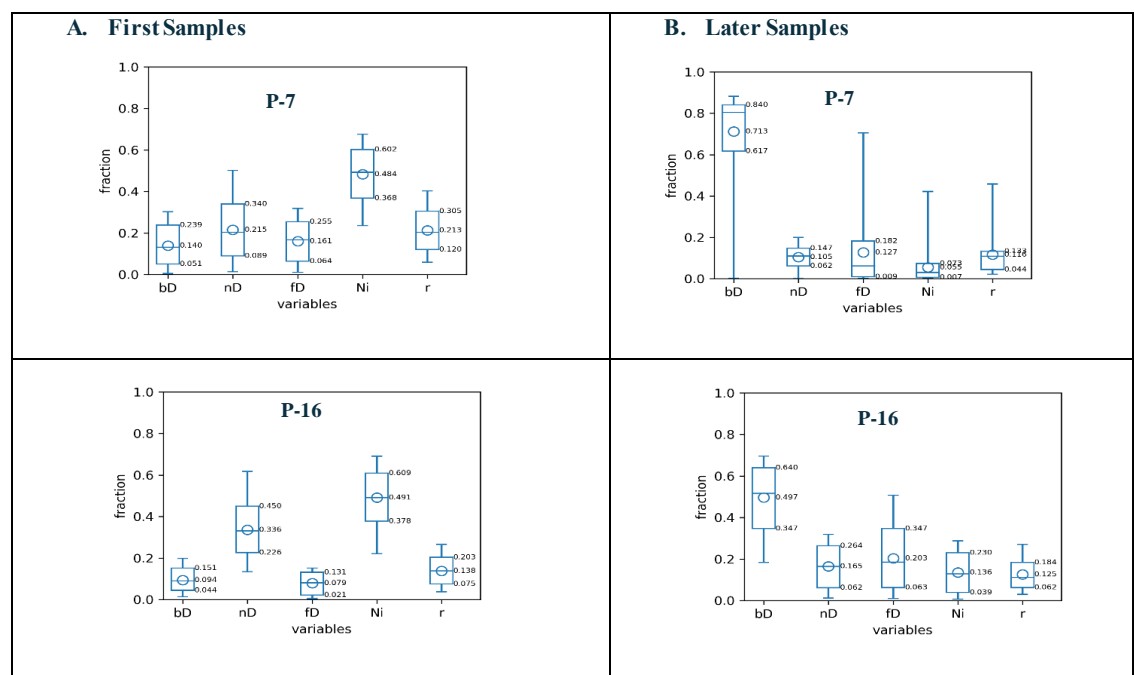



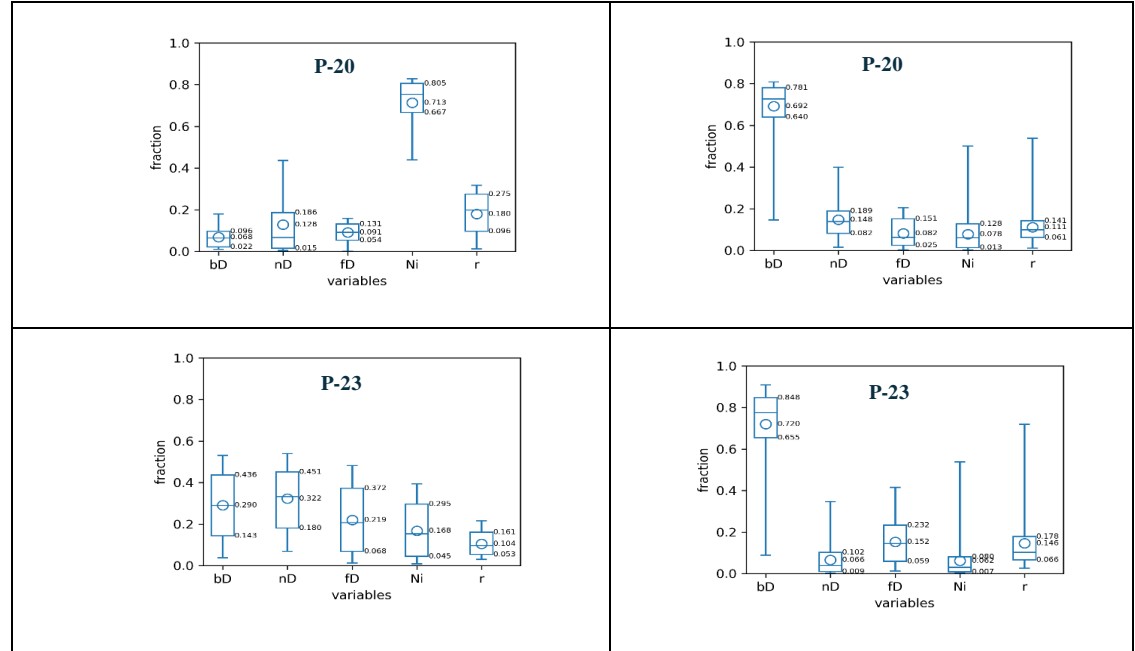

**Figure 8: N$_2$O production pathways contribution and N$_2$O reduction progress based on the FRAME modelling of the experimental N$_2$O isotope data collected for the first analysed samples (before glucose addition, 1$^{st}$ and 2$^{nd}$ sampling, day 2 and 4) (A) and mean value of the later samples (after glucose addition, 4$^{th}$ and 5$^{th}$ sampling, day 9 and 11) (B).** The estimated contributions of bacterial denitrification (bD), nitrifier denitrification (nD), fungal denitrification (fD), and autotrophic nitrification (Ni) illustrate the dynamic shifts in microbial pathways and N$_2$O reduction progress (r) over time.

The graphs, analyzed using the FRAME model, reveal distinct microbial processes driving N$_2$O production and reduction during laboratory incubation of groundwater samples from piezometers P-7, P-16, P-20, and P-23, comparing the initial incubation phase (1–2 days) to the later phase (4–14 days), including samples before and after glucose addition.

This division of samples was made after the observed isotopic signatures – the initial samples (day 2) showed no $^{15}$N enrichment in the N$_2$O and later samples (day 7 – day 14) were characterised with very significant $^{15}$N enrichment. Initially, autotrophic nitrification (Ni) dominated across all samples, contributing around 60–70% to N$_2$O production, while bacterial denitrification (bD) was lower, ranging between 20–30%. In P-7 and P-16, minor contributions from nitrifier denitrification (nD) (10–20%) and fungal denitrification (fD) (<10%) were observed, with similar trends in P-20 and P-23, where nD accounted for slightly higher fractions of N$_2$O production. Residual N$_2$O fractions (r$_{N2O}$) across all piezometers ranged between 10–26%, reflecting high partial N$_2$O reduction to N$_2$.

After glucose addition, microbial activity shifted significantly toward denitrification, with bD becoming the dominant pathway (up to 80–85%), driven by the availability of carbon. P-7 and P-





16 exhibited a gradual rise in bD, reaching up to 73%, while residual Ni contributions declined
correspondingly. In P-20 and P-23, the transition was sharper, with bD dominance occurring more
abruptly. Residual $N_2O$ fractions decreased across all samples as bD activity intensified.
Simultaneously, Ni contributions dropped below 10% in all samples, while nD and fD remained
minimal, contributing <15% to $N_2O$ production. Interestingly, for the last sample (day 14 of the
incubation) for all the analysed waters the model could not find any solution. This might be due
to accumulation of very different pathways contribution and progressing reduction of $N_2O$
originating from the mixture of all production pathways. Figure 9 presents illustration of the model
performance on an example of sample 6 (day 14) of the P-16, while this is similar for all the
piezometers. The modelling problem occurs due to too high $\delta^{18}O$ measured values, while $\delta^{15}N$ and
$\delta^{15}N^{SP}$ show values typical for bD, $\delta^{18}O$ is shifted to much higher values, indicating large
reduction, not confirmed with low $\delta^{15}N^{SP}$ values. This can result from the actual smaller O-isotope
exchange with water than the one assumed by for bD in the model input values. The endmember
values for bD are mostly determined based on soil experimental studies (Yu et al., 2020), hence it
is theoretically possible that slightly different range of values should be assumed for groundwater
studies. Another explanation could be significant admixture of chemodenitrification, which is
characterized by high $\delta^{18}O$ values ((Wei et al., 2019). This assumption might be supported by the
fact that quite significant $N_2O$ production was found in some of sterile samples, with especially
high production at the end of the experiment (Table 2). This $N_2O$ produced from sterile treatments
shows always high $\delta^{18}O$ values and very variable $\delta^{15}N^{SP}$ values (Fig. 5A).
This interpretation is further supported by the elevated $\delta^{18}O$ values in later incubation stages,
suggesting abiotic processes like chemodenitrification which may have contributed to $N_2O$
production. Chemodenitrification has been shown to produce $N_2O$ with distinct isotope
fractionation patterns, including elevated $\delta^{18}O$ values compared to microbial pathways (Chen et
al., 2021). The detection of $N_2O$ in sterile samples also points to a possible non-biological
contribution, as nitrite can undergo chemical reduction in the absence of microbial activity (Heil
et al., 2016). Furthermore, abiotic $N_2O$ formation has been linked to Fe(II)-mediated nitrite
reduction, particularly under anoxic conditions, with organic matter, including humic and fulvic
acids, potentially facilitating $N_2O$ production through chemical pathways (Zhu-Barker et al.,
2015). However, since Fe(II) presence in our sterile samples is unknown, other abiotic
mechanisms, such as organic matter interactions, cannot be ruled out.
Importantly, the FRAME model does not include chemodenitrification, which is most probably
the reason for biased results for the last samples. The discrepancies between modeled and observed
isotope values suggest that additional abiotic pathways, such as chemodenitrification, may need to
be considered in future isotope models to improve accuracy.





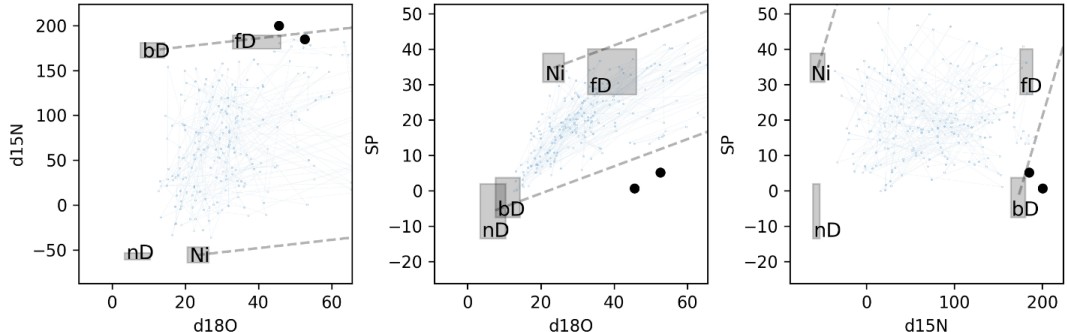

**Figure 9: An example of the FRAME modelling path illustration for the last incubation**
**samples (P-16, day 14), which do not provide modelling results.** The black dots illustrate
measured samples and the blue points the model Monte Carlo sampling. No coherence of measured
and modelled data indicate that the model did not find plausible solution for the given data.
All piezometers displayed a similar transition from nitrification-driven processes in the first
samples to denitrification-dominated processes in the later incubation days. However, at the final
sampling points, no fitted solution could be obtained for some data, suggesting the presence of
unknown processes or a complex overlap of microbial pathways. These findings indicate very
dynamic N$_2$O production processes while highlighting limitations in resolving mixed nitrogen
pathways at later stages. Moreover, while microbial denitrification was the primary N$_2$O source,
the observed discrepancies suggest that abiotic contributions, such as chemodenitrification, may
have been a more relevant factor than initially expected, particularly under conditions favoring
nitrite accumulation.
**4.2.3 Gene Abundance Shifts in Microbial Communities**
Pre-incubation data indicate a notable presence of aarchaeal *amoA* genes compared to bacterial
*amoA,* suggesting active archaeal ammonia oxidation in the samples (Fig. 6A). While
denitrification gene *nosZI*—show relatively high abundance in some samples (e.g., P-16 and P-
23), the consistent presence of archaeal *amoA* and the lower abundance of other denitrification-
related genes (*nirK, nirS,* ), , suggests nitrification processes were prominent prior to incubation.
This is particularly evident in P-7 and P-20, where archaeal *amoA* surpasses denitrification genes,
suggesting a stronger nitrification potential.
This is consistent with findings from (Mosley et al., 2022), which reported that ammonia-oxidizing
archaea (AOA) tend to dominate in oligotrophic groundwater environments with low ammonia
concentrations due to their higher affinity for ammonia and oxygen limitation, often outnumbering
ammonia-oxidizing bacteria (AOB). Similarly, the functional gene proportion analysis (Fig. 6B)
highlight the contribution of archaeal *amoA* genes to total prokaryotic abundance, emphasizing
their critical role in ammonia oxidation.. In contrast, the low proportions of bacterial *amoA* further
confirm limited bacterial involvement in N cycling prior to incubation. This has also been observed
in groundwater systems where bacterial nitrification potential remains constrained due to
environmental limitations on AOB populations. Post-incubation, there was a significant increase



in the abundance of denitrification genes like *nosZI*, partICularly in samples P-7, P-16 and P-20,
illustrating a shift from nitrification to denitrification under incubation conditions Fig. 6(A). This
aligns with (Wang et al., 2022), who found that site-specific environmental conditions, particularly
carbon and N availability drive microbial community shifts in N cycling, with increased
denitrification gene abundance. Functional gene proportions also reveal a corresponding rise in the
relative abundance of *nosZI*, illustrating the shift in microbial community function towards
denitrification processes Fig. 6(B). The abundance of DNRA and commamox genes showed
minimal changes, suggesting no difference in the presence of these processes between pre- and
post-incubation conditions. This observation is supported by (Broman et al., 2021), who reported
that DNRA gene abundance remained stable under experimental conditions, indicating its potential
resilience to shifts in N cycling pathway. The abundance of archaeal 16S rRNA genes decreased
in samples P-7 and P-23, indicating a reduction in archaeal community, whereas the abundance of
bacterial 16S rRNA increased significantly in P-16 and P-20, reflecting bacterial growth during
incubation. Paired T-tests confirmed these observations, showing significant increases in the
abundance of *nosZI* genes in P7 ($p<0.05$) and P16 ($p<0.05$) and shifts in the archaeal 16S rRNA
abundance ($p<0.05$), but not in the abundances of *nirK* or *nosZII* genes ($p>0.05$), highlighting the
variability in microbial responses. These results suggest that specific environmental or
experimental conditions during incubation can significantly influence certain microbial processes,
particularly those related to N cycling. This is consistent with (Wang et al., 2022), who found N-
cycling gene abundance varies with environmental factors like carbon and N availability.
Similarly, (Mosley et al., 2022) reported persistent transcriptional activity in nitrification and
denitrification across groundwater conditions, indicating microbial adaptability. The significant
results for the community of *nosZI* and archaeal 16s rRNA highlight their potential roles in
environmental monitoring and microbial ecology studies.

**796 4.2.4 The identification of active N transformations in the laboratory incubations**

The interpretation of the presented results is quite challenging, since this is the first study
combining the N and O isotope analyses of $NO_3^-$ and $NO_2^-$ as well as $N_2O$ isotopes including three
signatures: $\delta^{15}N^{SP}_{N2O}$, $\delta^{18}O_{N2O}$ and $\delta^{15}N_{N2O}$. The overall summary of this data is rather surprising
and may seem inconsistent, because the low-level $^{15}N$ label added to the $NO_3^-$ pool is not found in
the $NO_2^-$ pool, but almost completely transferred to the $N_2O$ pool. Both the $N_2O$ isotope results
and gene copy numbers document occurrence of intensive denitrification, especially in the second
phase of the incubation, whereas the analyses of inorganic N indicate simultaneous intensive
nitrification processes, with significant formation of $NO_2^-$ It is surprising due to very low levels of
$NH_4^+$ during the whole incubation and indicates that the additional unlabelled N must originate
from organic nitrogen pool (DON).
The FRAME analysis of $N_2O$ isotope data, T-test results, and gene abundance graphs together
show a shift from nitrification to denitrification in microbes during incubation, influenced by
adding carbon source and created suboxic conditions. Initially, the FRAME results show that
nitrification, mainly due to archaeal community (as seen with high levels of the archaeal *amoA*



gene), is the dominant N₂O production pathway. Isotope analysis supports this, with N₂O isotope
signatures characteristic for nitrification for the first samples (Fig. 5, Fig.8). Pre-incubation data
indicate that archaeal ammonia oxidation was a dominant process in samples P-7 and P-20, as
evidenced by higher archaeal *amoA* gene abundance relative to denitrification-related genes.
However, in samples such as P-16 and P-23, the abundance of *nosZI* suggests that denitrification
processes were also active, pointing to a co-occurrence of nitrification and denitrification processes
across the groundwater samples.

Post-incubation, FRAME results show an increase in bacterial denitrification (bD) fractions,
correlating with the significant rise in denitrification-related genes, particularly *nosZI* validated by
paired T-tests ($p<0.05$). These changes are confirmed by gene abundance graphs that show a
notable increase in these denitrification genes after incubation. The total prokaryotic abundance
also increased in P-16 and P-20, reflecting enhanced bacterial growth, whereas smaller changes in
P-7 and P-23 suggest variable responses to carbon addition (Fig. 6). A decline in nitrification genes
align with the FRAME-predicted reduction in nitrification activity. Additionally, isotopic data
revealed significant N₂O reduction to N₂ in most samples, consistent with bacterial denitrification
dominance, reduced contributions from nitrification pathways, and increase in the abundance of
genes responsible for N₂O reduction to N₂ (*nosZII*). Together, these results confirm microbial
transition from archaeal-driven nitrification to bacterial denitrification, highlight the role of carbon
availability and suboxic conditions in regulating N cycling. The integration of gene abundance,
isotope dynamics, and FRAME analysis provides a comprehensive understanding of the microbial
processes driving N transformations during incubation.

Importantly, the overall results showed that both reduction and oxidation processes are occurring
simultaneously in the studied aquifer. Theoretically, in our incubations the suboxic conditions
should rather favor denitrification $NO_3^-$ reduction. Indeed, the majority of the released N₂O is
formed due to bacterial denitrification from $NO_3^-$ as a substrate. However, $NO_2^-$ originate in large
majority from organic N oxidation with very minor fraction originating from $NO_3^-$ reduction.

**5. Conclusions and outlook**

This study demonstrates the intricate dynamics of N transformations in groundwater samples by
integrating isotope analyses, microbial gene abundance, and FRAME modeling to elucidate the
microbial mechanisms involved. Application of multi-compound isotope studies ($NO_3^-$, $NO_2^-$,
N₂O) combined with the novel idea of low-level $^{15}$N labelling and microbiome studies provide a
very detailed insight into the occurring processes and reveal some unexpected mechanisms. Based
on this complex dataset we can document the co-occurrence of the oxidation and reduction
pathways and existence of different, separated $NO_2^-$ pools.

$NO_2^-$ production is likely driven by nitrification processes linked to the oxidation of organic N
from the elevated DON levels in water samples. Also, the data indicated the simultaneous
occurrence of denitrification processes, particularly under suboxic conditions induced during



incubation, highlighting the dynamic nature of nitrogen cycling. Future investigations into the role
of DON could deepen understanding of its impact on nitrification and denitrification in waters.
Broader application of these integrated methods combining isotope analyses and microbial gene
studies in field-scale studies can improve monitoring and management of nitrogen pollution in
groundwater systems.
**6. Data availability**
Original data are available upon request. Material necessary for this study's findings is presented
in the paper.
**7. Author contribution**
Conceptualization was led by SD, with supervision from DLS and ME. Visualization (figures and
plots) prepared by SD and DLS. Microbiological analyses were conducted by SD and ME. SD,
DLS contributed to writing, methodology, investigation, data curation. Fieldwork and sample
collection were carried out by SD, DLS, MB, and MJ. Funding acquisition and resources were
supported by SD, DLS, ME and UM. Gas and isotope analyses were performed by SD, DLS, and
RW. We thank all our co-authors for their valuable support and feedback.
**8. Competing interest**
The authors declare that they have no conflict of interest.
**8. Financial support**
This study was financially supported by the "Polish Returns" programme of the Polish National
Agency of Academic Exchange and the grants Opus-516204 (PI: Dominika Lewicka-Szczebak)
624 and Preludium-522855 (PI: Sushmita Deb) of the National Science Centre Poland. Also,
supported by the Estonian Research Council (PRG2032), by the European Union Horizon program
under grant agreement No 101079192 (MLTOM23003R), and the European Research Council
(ERC) under grant agreement No 101096403 (MLTOM23415R).







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
