# Peer review of "N-transformations in nitrate-rich groundwaters: combined isotope and microbial approach"

_EGUsphere, 2025_

## Author Response (AR1)

**Authors responses to reviewers' comments**

RC1: 'Comment on egusphere-2025-754', Anonymous Referee #1, 27 Mar 2025

**Publisher's note: this comment was edited on 31 March 2025. The following text is not identical to the original comment, but the adjustments were minor without effect on the scientific meaning.**

Major Comments

One of the highlights of the manuscript "Enhanced isotopic approach combined with microbiological analyses for more precise distinction of various N-transformation processes in contaminated aquifer – a groundwater incubation study" is the simultaneous use of isotopic labeling and natural abundance stable isotope methods to investigate the nitrogen cycling processes in groundwater and the sources of nitrous oxide ($N_2O$) production. The study found that nitrite (NO2-) may originate from the mineralization of dissolved organic nitrogen (DON) under conditions of high dissolved organic nitrogen. Furthermore, the nitrogen transformation processes are highly sensitive to the availability of organic carbon. The increase in the C/N ratio due to the additional organic carbon promotes the transition of nitrification processes to denitrification processes in groundwater. This manuscript is significant for a deeper understanding of the sources of nitrogen pollution in groundwater and the migration and transformation processes of nitrogen. The conclusions of this study provide theoretical support for explaining the sources and mechanisms of high nitrate pollution in groundwater.

Thank you for your positive evaluation of our study and all the comments which helped us to further improve this work. We have addressed all the critical comments as explained below.

Specific comments

1. Line 45-48: It is suggested to explain the specific anthropogenic sources and natural processes involved in groundwater.

   Thank you for your comment. We have revised the manuscript and added the lines to the introduction:

Controlling $NO_3^-$ levels in aquatic systems presents substantial environmental challenges, particularly in groundwater, due to the complexity of differentiating between its anthropogenic sources- such as fertilizer runoff, waste from livestock manure, industrial

wastewater discharges and natural processes including soil organic matter mineralization, precipitation and biological nitrogen fixation by microorganisms.

2. Line 138-139: Supplement the salinity and dissolved oxygen concentration values of the groundwater samples collected from the monitoring wells.

Thank you for your comment. The dissolved oxygen (DO) concentration range of 2.2–4.3 mg L$^{-1}$ provided in the manuscript reflects typical suboxic groundwater conditions reported for this aquifer system and is based on previous field campaigns in the study area (Olichwer et al., 2012). In the present study, individual DO values were not systematically measured for each piezometer during sampling, and therefore, this range is intended to provide a general reference for groundwater conditions. We are aware that this is a major shortcoming of our dataset, but unfortunately this was due to unexpected problems. We did the sampling campaign in collaboration with routine monitoring action performed by private company, who should have measured all the physicochemical parameters. Only afterwards we discovered that O2 was not included and due to major logistics problems this could not be repeated. But from the archival data we see quite a narrow and stable range of these values, so we are convinced that this range is well representative for our current samples.

This detailed explanation will be also added in the manuscript.

3. Line 139-142: Why were these four monitoring well water samples selected for laboratory cultivation? Please clarify the selection criteria.

The major criterium was the nitrate concentration, but the second one was also the possible water gain from a piezometer. Some of the wells were quickly pumped out and no more water could be collected. This was eg. the case for P-0 and P-3 which exhibited comparably high nitrate concentration levels, but must have been excluded from the incubation experiments due to limited water availability at the time of sampling.

This detailed explanation will be also added in the manuscript.

4. Line 181-188: Indicate the detection limits for the inorganic nitrogen analysis methods.

For our analysis, wavelengths of 520 nm, 560 nm, and 610 nm were selected for $NO_3^-$, $NO_2^-$, and $NH_4^+$ concentration, respectively, with detection limits of 0.1–50.0 mg L-1 for $NO_3^-$, 0.02–1.0 mg L-1 for $NO_2^-$, and 0.01–5.00 mg L-1 for $NH_4^+$.

This information was added in the manuscript.

5. Line 204-207: What is the spatial relationship between these monitoring wells and the discharge point (the retention area for yeast production wastewater)? Is there a possibility that the high DON in the yeast wastewater mineralizes to ammonium and subsequently nitrifies to NO3-? Table 1 shows that the monitoring wells with high NO3- concentrations also have higher DON concentrations (except for the discharge point P-L1). If this is the case, it is necessary to also focus on monitoring wells with high ammonium and low NO3- to compare with high NO3- water samples, thus revealing the process of high DON converting to NO3- in groundwater and the nitrogen transformation conditions.

Thank you for this important comment. Yes, the process is undergoing as you described. We have revised the manuscript and added the following paragraph in the discussion section to address this point:

'The increased concentrations of both DON and $NO_3^-$ in most piezometers suggest that organic nitrogen input may contribute to higher $NO_3^-$ concentration. Although direct groundwater flow paths were not explicitly studied, the spatial positioning of these wells to the lagoon, along with their elevated DON levels, supports the possibility of influence from wastewater discharge. The precise knowledge of the δ15N signature of the potential N substrates, i.e. of DON and waste waters, could further confirm the dominant source of the samples (Boumaiza et al., 2024).

Interestingly, the $NH_4^+$ content is very low in the piezometers of high $NO_3^-$ content (Tab. S1) indicating its rapid nitrification. Most possible explanation suggests microbial mineralization of DON to $NH_4^+$, with instantaneous rapid nitrification to $NO_3^-$ under suboxic conditions. A few piezometers with elevated $NH_4^+$ content show very low $NO2^-$ and $NO_3^-$ contents, which may suggest that the nitrification processes are not active there. However, these waters were not selected for further incubation studies, due to our focus on $NO_3^-$ formation and the selection of $NO_3^-$-rich waters. Future studies should integrate groundwater level measurements or tracer-based studies to confirm source connectivity between lagoons and piezometers, along with sampling of $NH_4^+$-rich, $NO_3^-$-poor locations for better analysis of nitrogen transformation pathways.'

,

6. Line 213-215: Does this mean that the DO concentration in the water of the monitoring wells falls within this range? Since this data is not shown in Table 2, it is suggested to include the DO concentration values of the field monitoring wells in Table 1.

Thank you for your comment. The dissolved oxygen (DO) concentration range of 2.2–4.3 mg L$^{-1}$ provided in the manuscript reflects typical suboxic groundwater conditions reported for this aquifer system based on previous field campaigns in the Wołczyn region (Olichwer et al., 2012). In the present study, individual DO values were unfortunately not systematically measured for each piezometer during sampling, and therefore, this range is intended to provide a general reference for groundwater conditions.

We have clarified this point in the revised manuscript to avoid confusion with the DO values reported in Table 2, which reflect dissolved measurements dynamics during laboratory incubation, at different time points.

7. Line 220-221: Why is the volume of the added labeled liquid different here?

The added volume of $^{15}$N-labelled tracer solution was adjusted according to the nitrate concentration in each sample to achieve a comparable level of isotopic enrichment across all samples while minimizing alteration to the natural isotopic composition.

8. Line 263-266: Please specify the detection limit for the denitrifying bacteria method.

The denitrifier method using Pseudomonas aureofaciens enabled isotope analysis of NO3- at concentrations as low as 40 nmol NO3− L-1 (Stock et al., 2021) (Deb and Lewicka-Szczebak 2024), while using Stenotrophomonas nitritireducens allowed for NO2- analysis at concentrations as low as 150 nmol NO2− L-1 (Deb and Lewicka-Szczebak, 2025).

This information was added in the manuscript.

9. Line 308-309: The NO3- concentration in the P-0 monitoring well is higher than that in P-7, yet it was not selected for further cultivation studies. Why?

Although P-0 and P-3 also exhibited comparably high nitrate concentration levels, they were could not be further studied in the incubation experiments due to limited water availability at the time of sampling.

This clarification was added in the manuscript:

'Due to restricted water availability in some piezometers, where the aquifer was quickly pumped out and the water amount required for the further incubations (min. 750 mL) could not be collected, some potentially interesting piezometers must have been omitted. This was the case for eg. P-0 and P-3 where despite high NO3-concentrations the incubations were not possible due to too less water gain.'

10. Line 314-316: Please add the detection method for NH4+ isotopes and its detection limit in the methods section.

Thank you for the comment. We would like to clarify that although $NH_4^+$ concentrations were measured in our study, stable isotope analysis $\delta^{15}N\text{-}NH_4^+$ was not performed due to concentrations being too low.

This clarification was added in the manuscript.

11. Line 600: Should it be "labelled NO3- reduction"?

Yes, thanks for the correction.

12. Line 632-633: According to Figure 7, before the addition of glucose, some points (despite their low NO2- concentrations) tended towards heterotrophic nitrification, while after the addition, they clustered more in the autotrophic nitrification range. Please verify the data and correct the result description.

Thank you for your comment, you are right. We have revised the discussion in the manuscript as follows:

'Before the addition of glucose, the isotopic signatures (empty circles) were more scattered with several points within or near the heterotrophic nitrification zone, suggesting a mix of both autotrophic and heterotrophic pathways to nitrite production under low-carbon conditions. . Some clustered points were also observed near autotrophic nitrification area indicating that autotrophic bacteria, such as Nitrosomonas europaea were likely involved in conversion of ammonia ($NH_3$) to nitrite ($NO_2^-$) as an energy-generating process (Deb et al., 2024). During this process, $CO_2$ serves as the sole carbon source for these bacteria, assimilated into their biomass to support cellular growth, independent of the chemical reaction used for energy generation (Hommes et al., 2003). In the groundwater samples, $CO_2$ likely originated from the decomposition of organic matter in the yeast sewage (Section 2.1) or from the carbonate system naturally present in groundwater (Section 3.1.1).

The application of yeast-based sewage as fertilizer in the agricultural site likely introduced organic nitrogen into the groundwater, which undergoes microbial decomposition to release ammonium (NH4+) through the mineralization of proteins and amino acids (Watanabe et al., 2023). However, NH4+ was not detected in the groundwater samples ( Table 1), indicating its rapid transformation within the nitrogen cycle, such as nitrification and assimilation.. Microbial assimilation likely contributed to NH4+ exhaustion, as microbes utilized it for biomass synthesis. However, this is not the case for sterile

samples, where we observe slight accumulation of $NH_4^+$, indicative of biological uptake in $NH_4^+$ turnover.

Following glucose addition, the isotopic signatures (filled circles) were more concentrated within the autotrophic nitrification zone, which indicates that autotrophic nitrification continued to dominate nitrite production despite under elevated carbon conditions. While a shift towards heterotrophic nitrification might be expected under increased carbon availability, the isotope data suggest that autotrophic ammonia-oxidizing bacteria remained more metabolically active than the heterotrophs under the given incubation conditions. Together, these findings demonstrate the rapid transformation of NH4+ from yeast-based fertilizers into intermediate nitrogen compounds, driving nitrification and subsequent nitrogen cycling processes in groundwater'.

'

13. Line 804-806: This conclusion is very interesting. Was there simultaneous monitoring of DON during the cultivation period? What was the magnitude of the DON changes?

Thank you for this comment.

Dissolved organic nitrogen (DON) concentrations were determined in the initial groundwater samples prior to incubation and are presented in Table 1, but variation in DON during the incubation period were not assessed. We admit it should have been done, but at the time of performing the incubations we did not really expect such an important role of DON mineralization. We constructed this incubation experiment rather to check denitrification rates and the associated isotope effects. Only when evaluating the results, which were partially very surprising, we discovered the importance of DON unusually high concentration, as for groundwaters. Therefore, we based our conclusion about DON's role in nitrite production on the high initial DON concentrations and also the isotopic data, which suggest that nitrogen from natural (unlabelled) sources — likely including DON — was actively involved in the nitrification process.

14. Line 845-846: The samples collected in this study are all high in NO3- or DON, and samples with high NH4+ concentrations were not analyzed. Therefore, the sources of NO2- under high ammonium conditions and the relationship between NO2- and DON at that time cannot be shown. This is a significant limitation of the study. It is suggested to address this in the discussion.

Thank you for this comment. We have revised the manuscript and added this point to the discussion.

'While this study focused on samples with elevated $NO_3^-$/DON concentrations to investigate nitrogen transformation processes, samples with higher $NH_4^+$ concentrations were not included in the incubation experiments as they typically showed low $NO_3^-$ levels and below detection isotope signatures, further limiting their utility for isotopic-based analysis. As such, the role of $NH_4^+$ in $NO_2^-$ formation under such conditions could not be evaluated and requires further research'.

**Citation**: https://doi.org/10.5194/egusphere-2025-754-RC1

**RC2**: 'Comment on egusphere-2025-754', Anonymous Referee #2, 27 Apr 2025

The work by Deb et al. presents an isotope study of inorganic N and N2O turnover from a contaminated groundwater system. With both natural abundance and light 15N-tracing methods, they determined the source/process contributions based on empirical isotope fractionation effects. In addition, by adding glucose to several groundwater samples, they examined the response in N turnover processes. Overall, this is a detailed mechanistic work exploring nitrate, nitrite and N-gas production dynamics from groundwater system. However, much remains to be improved and clarified.

Thank you for your positive evaluation of our study and all the comments which helped us to further improve this work. We have addressed all the critical comments as explained below.

- This work is written in a loose manner, and organized in a sophisticated way. For example, lots of "raw" data were presented in a simple table without better organization and comparison. A lot of paragraphs include only one or two sentences. For sections 4.1 and 4.2, results/discussions were simply piled up without better sorting for highlighting important outcomes. Further, almost half of the figures and tables, together with descriptive contents for methods, should be moved to SI.

  Thank you for this comment. We have revised the manuscript – we deeply reorganized the sections and improved the clarity. We moved 3 tables and 2 figures to the SI and clearly divided results and discussion, to keep batter focus on important results through the discussion section.

- The closed-system assumption is very important for such source partitioning work based empirical isotope fractionation effects. While the key assumption is not too different from previous work, the major uncertainties still exist. It is important to limit the discussion of source contributions, and provide some uncertainty discussions.

  Thank you for this comment. We have revised the manuscript. Discussion on this issue was added:

  'Isotope-based source partitioning in this study assumes a closed system approach. However natural groundwater environments may often exhibit open-system behavior due to water movement, nutrient inflow, and microbial activity and therefore, the estimated source contributions- especially based on isotope mass balances and FRAME modeling may represent results under controlled laboratory conditions. Although tracer application and microbial data helped minimize uncertainties from concurrent processes such as DON oxidation or oxygen exchange, certain limitations still apply and require future field applications to validate closed-system approaches under varying conditions.'

- The water-incubation experiments were somewhat disconnected with the real-field scenarios, at least based on the current manuscript version. The authors should further rationalize why glucose addition is important. Also, how would the knowledge of the incubation study be transferred to real-field understanding?

  Thank you for this comment. We have revised the manuscript by adding the links between field and laboratory studies and additional explanation of glucose application was added as follows:

  'To simulate microbial denitrification under suboxic conditions, glucose was added as a carbon source. This approach is also supported by previous research (Liu et al., 2022) where external carbon addition, particularly glucose, can significantly enhance biological denitrification and nitrate removal efficiency in groundwater. This helps understand how elevated organic carbon (e.g., from wastewater or agricultural leachate) could influence N-transformations in the field. In our study, the observed shift from archaeal-driven nitrification to bacterial denitrification highlights the role of carbon availability in nitrogen cycling pathways. While lab incubations cannot fully mimic complex field-scale conditions, they provide insights into microbially mediated processes. Future in situ studies incorporating natural carbon amendments would help validate these findings under real, open-system aquifer conditions'

Title: the current title is too long. In addition, "enhanced" may not fully represent the research approach here and it is difficult to define which is more "enhanced". Considering that microbial analysis and incubation have been applied, why not call this approach a "combined" or something similar?

Thank you for this important suggestion. The title has been changed to shorter and clearer version: 'N-transformations in nitrate-rich groundwaters tracked with combined isotope and microbial approach'

Line 50-56: While NO3- and NO2- are two major components in the groundwater environment, some more literature review regarding inorganic N (particularly NO2-) pollution and related turnover process should be mentioned.

Thank you for this comment. We have revised the manuscript

'Also, Nitrite ($NO_2^-$), often a transient intermediate in the nitrogen cycle can accumulate under certain environmental conditions and pose significant environmental and health risks such as methemoglobinemia (blue baby syndrome) in infants and carcinogenic nitrosamines (Ward et al., 2018). Further, elevated $NO_2^-$ levels in water bodies and agricultural soils, can be toxic to aquatic life and humans, and therefore underscore the importance of monitoring $NO_2^-$ alongside $NO_3^-$ in groundwater systems.'

Line 240-250: What were the calibrate standards used in both this study and in the comparison at Thünen Institute? How many calibration points were available?

We applied the recently produced and distributed standards in frame of DASIM project (DASIM 16, DASIM 17, 1ppm N2O) (Well et al., 2024). In both laboratories two standards have been used with high and low N2O isotope values, so that isotope normalization of each isotope value is performed based on 2 point calibration line. Additionally, as a quality check standard the commercially available technical N2O gas mixture was applied – to check for stability, drifts and calibration control. Additionally, the set of different N2O concentration standards (produced from technical N2O gas bottle) was applied to perform the amount correction, especially needed for CDRS analysis.

A clarification on the standards applied has been added into the manuscript.

Line 316-318: To indicate denitrification process, the NO3- concentration also matters. According to Kendall et al. (2007), the reduction of NO3- concentration should coincide with the increase of 15N-NO3-, which allows the estimation of isotope enrichment factor.

Thank you for this comment. We have added this information with the respective citation in the manuscript

' According to (Kendall et al., 2007; Kendall and Aravena, 2000) a parallel decrease in $NO_3^-$ concentration and increase in $\delta^{15}N–NO_3^-$ is characteristic of denitrification and allows estimation of nitrogen isotope enrichment factor, and helps quantify microbial $NO_3^-$ reduction.'

Section 4.1: As for quite affirmative deductions of NO3- or NO2- sources/processes (e.g.,Line 461-463; Line 481-483; Line 502-504), it is important to note that such source partitioning may only apply to close-system assumption. In groundwater studies, one-time sampling may be rather uncertain and cannot fully avoid the uncertainties from water exchange and nutrient diffusion.

Thank you for this comment. We definitely agree that a close system assumption is not valid for groundwater filed studies and we added the critical discussion about this issue.

However, we believe that the deductions of NO2- sources and the relation NO3-NO2 gives a true insight into the ongoing processes, because NO2- is a very short-living compound and actually provides a snapshot of actually occurring processes, it can be really transported on the longer distance, so even in open system the formation of this compound (NO2-) most probably only depends on the in-situ available substrates.

We have revised the manuscript as follows:

'Although the isotope signatures provide strong evidence for active denitrification and mixed nitrification pathways in the aquifer, it is important to acknowledge that these isotope-based interpretation of $NO_3^-$ and $NO_2^-$ transformations are based on single-timepoint groundwater sampling in open aquifers. Therefore processes such as water exchange, nutrient diffusion, and variations in nitrogen transformation may also influence the observed isotopic signatures. To better understand which potential N transformations are active in the aquifer we performed the laboratory incubations under controlled conditions of the selected groundwater samples.'

Section 4.2: The part for 4.2.1 is too long and written in a complexed way. It is better to move lots of descriptive contents into the result section.

Thank you for this comment. We have revised this section by moving the results-descriptive parts to the results section and the Figures 8 and 9 into the SI.

Line 700-709: How to rationalize the difference among P-7, - 16, -20 and -23, regarding the source contributions after glucose addition?

Thank you for this valuable comment. We have revised the manuscript.

'The observed variations across the piezometers after glucose addition were not uniform and can be attributed to site-specific conditions such as initial nitrate concentrations, DON levels, and also variation in microbial communities. Notably, nosZI gene abundance increased across all piezometers, with higher enrichment in P-16 and P-20, and suggests enhanced denitrification potential. Conversely, archaeal amoA gene abundance declined—particularly in P-7 and P-23—indicating a microbial shift from archaeal-driven nitrification to bacterial denitrification. These patterns highlight how suboxic, carbon-rich conditions can selectively enhance denitrification, depending on environmental conditions.'

References"

Well, R., Buchen-Tschiskale, C., Burbank, J.,Dannenmann, M., Lewicka-Szczebak, D., Mohn, J., Rohe, L., Scheer, C., Tuzzeo, S. and Wolf, B., 2024. Production of standard gases for routine calibration of stable isotope ratios of N2 and N2O. EGU General Assembly 2024, EGU24-11996.

**Citation**: https://doi.org/10.5194/egusphere-2025-754-RC2